# *Trans*-eQTL hotspots shape complex traits by modulating cellular states

## Graphical abstract

## Authors

Kaushik Renganaath,
Frank Wolfgang Albert

## Correspondence

falbert@umn.edu

## In brief

Renganaath and Albert use data from a large cross between two yeast strains to examine the relationship between genetic influences on gene expression and on complex traits. They reveal a major role for *trans*-acting hotspots that affect the expression of numerous genes and alter core biological processes and cellular states.

## Highlights

- Integration of transcriptomes and complex traits in 1,000 recombined yeast strains

- Discovered thousands of genetic correlations between gene expression and growth

- *Trans*-acting regulatory hotspots drive trait heritability and genetic correlations

- Cell states balancing core biological processes are enriched in genetic correlations

Renganaath & Albert, 2025, Cell Genomics 5, 100873
May 14, 2025 © 2025 The Authors. Published by Elsevier Inc.

CellPress

# *Trans*-eQTL hotspots shape complex traits by modulating cellular states

Kaushik Renganaath[1,2] and Frank Wolfgang Albert[1,3,*]
[1]Department of Genetics, Cell Biology, and Development, University of Minnesota, Minneapolis, MN 55455, USA
[2]Present address: Department of Medicine, University of Chicago, Chicago, IL 60637, USA
[3]Lead contact
*Correspondence: falbert@umn.edu

## SUMMARY

Regulatory genetic variation shapes gene expression, providing an important mechanism connecting DNA variation and complex traits. The causal relationships between gene expression and complex traits remain poorly understood. Here, we integrated transcriptomes and 46 genetically complex growth traits in a large cross between two strains of the yeast *Saccharomyces cerevisiae*. We discovered thousands of genetic correlations between gene expression and growth, suggesting potential functional connections. Local regulatory variation was a minor source of these genetic correlations. Instead, genetic correlations tended to arise from multiple independent *trans*-acting regulatory loci. *Trans*-acting hotspots that affect the expression of numerous genes accounted for particularly large fractions of genetic growth variation and of genetic correlations between gene expression and growth. Genes with genetic correlations were enriched for similar biological processes across traits but with heterogeneous direction of effect. Our results reveal how *trans*-acting regulatory hotspots shape complex traits by altering cellular states.

## INTRODUCTION

Genetic variation among individuals shapes phenotypic variation, but the mechanistic basis of how DNA variants affect most traits remains elusive. The vast majority of genetic associations found by genome-wide association studies (GWASs) of human complex traits reside in non-coding DNA and are not in linkage disequilibrium (LD) with protein-coding exons.[1] The causal variants at these GWAS loci likely act by altering the expression of at least one gene whose abundance is critical for the given trait.[2] Catalogs of expression quantitative trait loci (eQTLs; regions of the genome that contain regulatory DNA variation) obtained in tissues from reference populations show enrichment of eQTLs at GWAS loci, and vice versa.[3–13] There is a growing list of validated causal connections between the expression of a specific gene and a complex trait in humans[14–16] and model organisms.[17–20]

Recent results have added nuance to straightforward models of how regulatory variation alters gene expression and complex traits. In particular, many GWAS loci in humans still lack colocalized eQTLs, even though reference catalogs from dozens of tissues contain eQTLs for nearly every gene.[21] Many known reference eQTLs, at genes likely to affect a given complex trait, do not colocalize with GWAS hits at the same genes.[22] The expression changes caused by known reference eQTLs mediate only a small fraction (on average 11%) of complex trait heritability.[23] These limited contributions of known reference eQTLs to complex traits may be due to systematic differences between the variants that are detectable in current human eQTL versus GWA studies.[24]

Here, we investigated the intersection of regulatory variation and complex traits by integrating published eQTLs and QTLs that shape complex organismal traits from two well-powered datasets in the yeast *Saccharomyces cerevisiae*[25,26] (Figure 1A). Both datasets were gathered in the same set of about 1,000 haploid, fully sequenced, recombinant progeny (segregants) obtained by crossing the laboratory strain BY (a close relative of the reference genome strain S288C) and the vineyard strain RM. These two strains differ at about 1 out of every 200 bp, providing a rich reservoir of genetic variation for the dissection of complex traits.

The eQTLs were mapped using RNA sequencing data from segregants growing exponentially in a common laboratory condition (yeast nitrogen base [YNB], medium without amino acids at 30°C). The dataset includes 36,498 eQTLs affecting the expression of 5,643 genes.[25] The eQTLs fall into two classes. First, more than half of the genes are influenced by one "local" eQTL per gene, which are typically caused by DNA variants in *cis*-regulatory elements in the gene's immediate vicinity.[27,28] Second, the majority of the eQTLs (92%; 33,529) were located far from the genes they influence, typically on a different chromosome. These *trans*-acting eQTLs affect the expression of their target gene via altered expression or activity of an intermediate, diffusible factor. In this cross, *trans*-eQTLs are clustered at 102 hotspot locations that each influence the expression of numerous genes. Similar *trans*-eQTL hotspots exist in other yeast crosses[29] and the wider yeast population[30]; in crosses of strains in other species, including plants,[31] nematodes,[32] and rodents[33]; and may also exist in human populations.[34–37] Hotspots

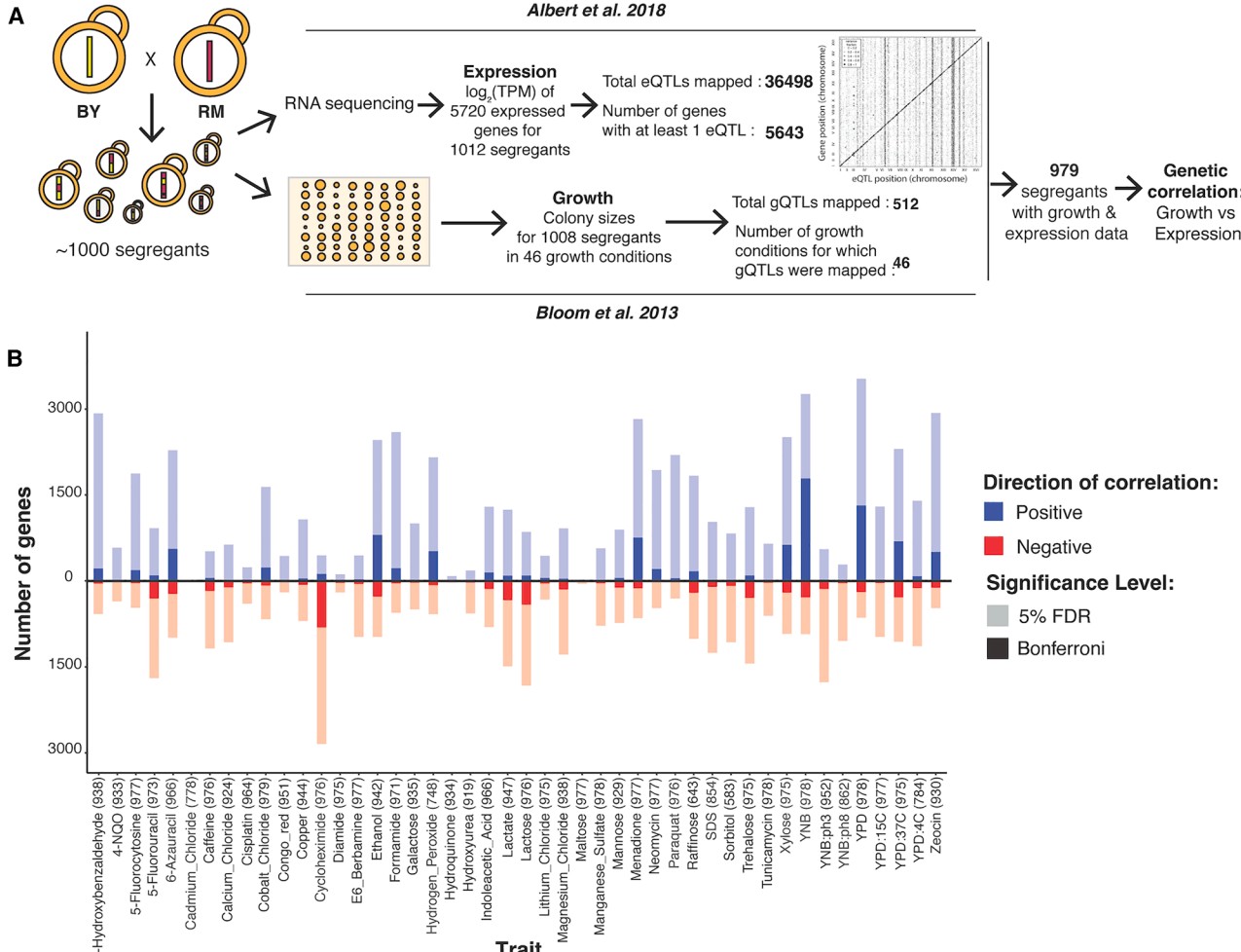

**Figure 1. Genetic correlations**

(A) The datasets used to compute genetic correlations.

(B) Number of genes with significant genetic correlation in each of the 46 growth conditions at 5% FDR (pale colors) and Bonferroni significance (saturated colors); both multiple-testing corrections performed within-trait. Positive correlations are in blue and above the zero line; negative correlations are in red and below. For each condition, the number of segregants with both growth and expression measurements in Bloom et al.[26] and Albert et al.[25] used to compute the genetic correlations are indicated in parentheses.

See also Figures S1–S6, Tables S1, S2, and S3.

in the BY/RM cross have been shown to arise from causal DNA variants with diverse molecular consequences in genes with a wide range of functions, from transcription factors to enzymes, and include lab-derived alleles as well as alleles that are common across *S. cerevisiae* isolates.[17,18,38,39]

As complex organismal traits, we used relative growth in a range of environmental conditions. Specifically, the segregants were grown on solid agar plates under one of 46 varied conditions, such as different temperatures or various chemicals. Growth was measured as colony size after a 2-day incubation. Genetic mapping yielded 591 QTLs across the 46 conditions.[26] Below, we refer to these QTLs as "growth QTLs" (gQTLs) and use the more general "QTLs" to encompass both eQTLs and gQTLs.

These datasets provide several key advantages. First, the identified QTLs account for a median of 72% (eQTLs) and 88%

(gQTLs) of narrow-sense heritability,[25,26] permitting analyses that capture most genetic variation in both gene expression and growth. Individual detected QTLs account for as little as ~1% of phenotypic variance, such that any undetected loci make at most modest contributions. Second, there are multiple eQTLs for nearly all genes (a median of 6 and up to 21; nearly all are *trans*-eQTLs) and multiple gQTLs for all traits (a median of 12, ranging from 5 to 29). This enables analyses across independent genetic loci across the genome. Third, gene expression and growth were measured in separate experiments conducted independently, years apart, at different institutions, and with segregants grouped into different batches, minimizing shared environmental factors between datasets. The gene expression data cannot have been influenced by exposure to the 46 conditions. Therefore, associations between gene expression and growth

observed across these datasets most likely arise from a shared genetic basis. In this regard, our design is analogous to comparisons between human eQTL reference panels from individuals without a given disease and GWAS results from individuals with a given disease. In contrast to these studies, our eQTLs and gQTLs were mapped in the same genotypes (for similar designs in model organisms, see references 31 and 40–46). Integration of these datasets promises to yield insights into how genetic variation in gene expression affects complex, organismal phenotypes.

Our analyses revealed thousands of significant correlations between the expression of individual genes and growth in a given condition. A prominent source of these genetic correlations was *trans*-eQTL hotspots that modulate central biological processes, reflected in expression change at hundreds of genes. Different traits showed positive or negative correlations with the same processes. These results shed light on the consequences of regulatory variation and serve as a resource connecting gene expression and complex growth traits.

## RESULTS

### The expression of thousands of genes is genetically correlated with growth in dozens of conditions

To identify associations between gene expression and genetically complex organismal traits, we tested for correlations between the expression of each of 5,643 genes that are affected by at least one eQTL in liquid YNB medium[25] and growth in 46 environmental conditions[26] (Figure 1A; Table S1). In principle, a correlation between two phenotypes (e.g., gene expression and growth) across datasets can be caused by environmental and genetic sources.[47] Given that our datasets were collected in independent experiments with numerous differences, environmentally induced correlations are unlikely in these data. Instead, any observed correlations likely arise from genetic variation that shapes both gene expression in the standard YNB condition and growth in the 46 environments.

Thousands of genes showed a significant correlation between expression level and growth in at least one condition at a false discovery rate (FDR; estimated as $q$ values within each trait) of 5% (median across traits: 2,296 range: 15 in cadmium chloride to 4,190 in YNB; Figure 1B; Tables S2 and S3). These thousands of genetic correlations per growth trait were robust to correction for shared factors among growth traits such as the solid agar medium to which conditions were added (Figures S1–S3), suggesting that they reflect trait-specific biology. The median absolute correlation coefficients of significant genetic correlations ranged from 0.09 (paraquat) to 0.18 (cadmium chloride), with a median across traits of 0.11 (Figure S4). These effect sizes suggest that genetic effects on gene expression in a standard environment shape growth in different environments partially but not completely, resembling recent results in humans.[23] Downsampling showed that detection of significant correlations of these magnitudes was possible due to the large size of the segregant panel (Figures S5 and S6). The thousands of genetic correlations revealed here are due to unknown sets of genomic loci that each influence both gene expression and growth. Below, we first identify

the responsible loci. We then explore biological processes involved in the genetic correlations.

### Most growth QTLs are colocalized with multiple local eQTLs

We first focused on local eQTLs, defined as eQTLs whose physical confidence intervals span the gene whose expression they affect (Figure 2A). More than half of all expressed genes in this cross have a detectable local eQTL.[25] Local eQTLs tend to be stronger than *trans*-eQTLs, intuitively suggesting that they could be a major source of genetic correlations. All but one of the 591 gQTLs in our dataset overlapped with at least one local eQTL, slightly but significantly more than expected by chance (median of 1,000 random gQTL sets: 586, $p = 0.023$). We used colocalization tests[51] to focus on 1,052 pairs of strong overlapping local eQTLs and gQTLs that are likely to share causal variants (Figure 2A; STAR Methods). Nearly all of the tested gQTLs (178/188; 95%; Figures 2B and 2C) had such a colocalized local eQTL, conservatively defined as QTL pairs at which colocalization was not rejected at a threshold of $p < 0.05$ without multiple testing correction (STAR Methods).

In a straightforward model of how regulatory variation influences a trait, a gQTL is due to a single strong local eQTL for a relevant gene. A likely example is a gQTL on chromosome X that affects growth in the presence of tunicamycin. This gQTL overlaps with three local eQTLs (Figure 2D), but only the eQTL for *CHS6* was consistent with shared causal variants. Deletion of *CHS6* leads to increased resistance to tunicamycin.[52] This observation is in line with the effects at this locus, where the RM allele decreased *CHS6* expression ($r = -0.24 \pm 0.03$) and increased growth in tunicamycin ($r = 0.2 \pm 0.03$).

In contrast to this straightforward model, 87% of tested gQTLs (163/188) were colocalized with multiple local eQTLs (a median of 5, and up to 22; Figure 2B; Table S4). Rather than suggesting causal genes that underlie the gQTL, many of these colocalizations are likely due to the linkage between gQTLs and the many local eQTLs in this one-generation cross (Figure 2A, bottom). Inspection of individual gQTLs with known causal genes confirmed this expectation (Figure S7). Furthermore, the magnitudes of genetic correlation at gene/trait pairs were not strongly dependent on whether a local eQTL for the gene was colocalized with a gQTL for the trait. While genetic correlations were stronger for gene/trait pairs where a local eQTL overlaps a gQTL than for such pairs with local eQTLs that did not overlap a gQTL (average absolute $r = 0.095$ versus 0.066; t test $p < 2.2e{-}16$), there was no difference in the strength of genetic correlations between gene/trait pairs with local eQTL/gQTL overlap that was classified as colocalized versus not colocalized ($p = 0.95$). There was also no association between colocalization status and the presence of a significant genetic correlation (Fisher's exact test [FET]: $p = 0.3$). A conservative interpretation of these results is that many, and perhaps most, local eQTLs are not causal for their colocalized gQTLs.

### Multiple *trans*-eQTLs per gene contribute to genetic correlations

Local eQTLs in a biparental cross represent the effects of linked variants in a single genomic region, effectively providing just a single datapoint for comparing genetic effects on gene expression

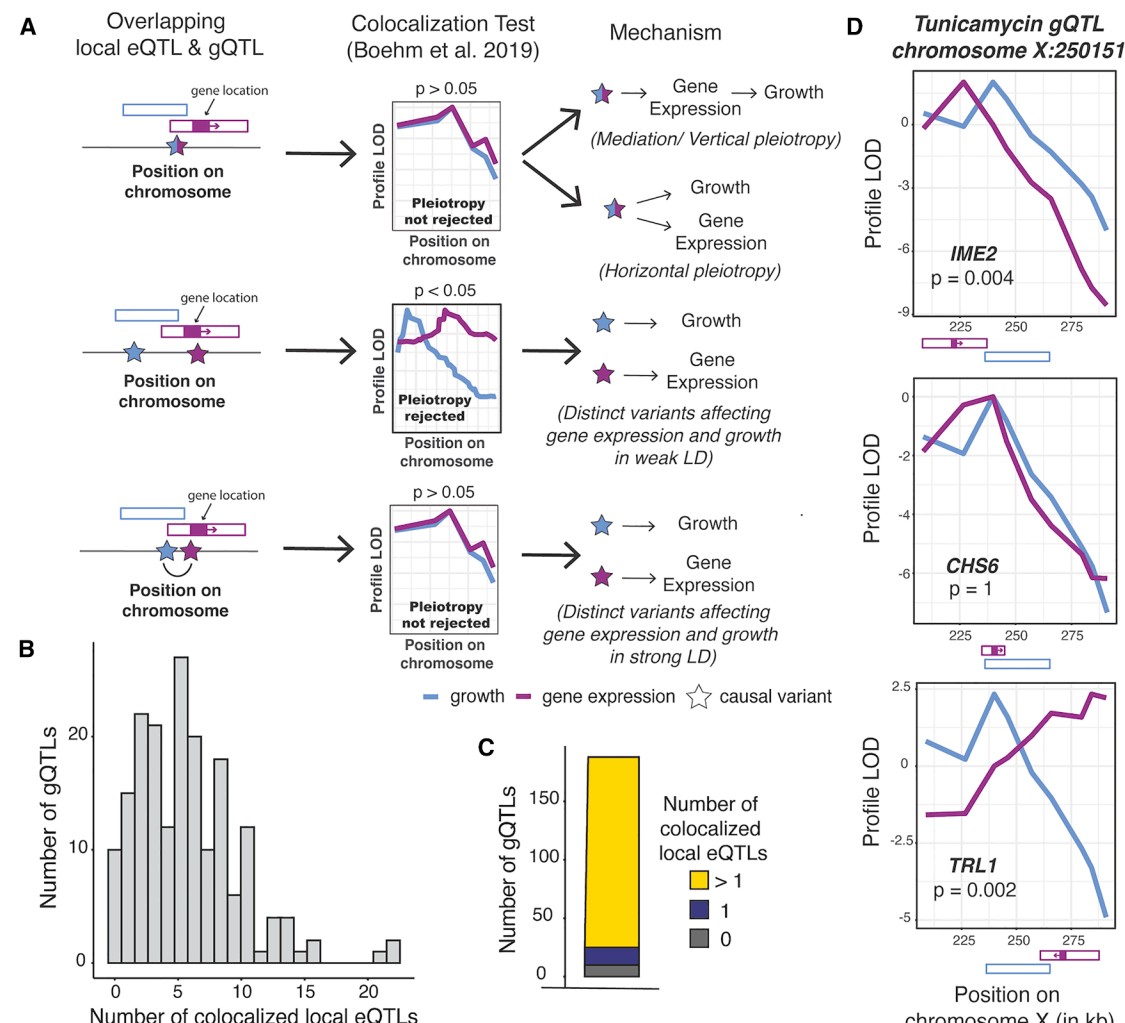

**Figure 2. Local eQTL analyses**

(A) Illustration of causal models at local eQTLs that overlap a gQTL. QTL locations are shown as hollow boxes; gene position and transcriptional direction is indicated as solid purple box and arrow. Under mediation or vertical pleiotropy,[48,49] causal DNA variants alter gene expression in the baseline condition, which then affects growth when segregants encounter the given environmental condition. Under horizontal pleiotropy, causal DNA variants also affect gene expression and growth but do so independently, through distinct pathways. Overlapping QTLs can also arise from distinct, linked causal variants in physical proximity that affect only gene expression or only growth. QTLs with shared, pleiotropic variants are called colocalized, to distinguish them from QTLs that overlap due to simple proximity between distinct causal variants.[50] To distinguish between these scenarios, we performed colocalization tests.[51] A significant *p* value on this test indicates that a single QTL caused by shared causal variants is rejected in favor of two separate QTLs with different causal variants.

(B) Histogram of the number of gQTLs with a given number of colocalized local eQTLs.

(C) The number of gQTLs with zero, one, and more than one colocalized local eQTL.

(D) Profile LOD curves for growth in tunicamycin (blue) and the expression of indicated genes (purple) at a gQTL on chromosome X. Profile LOD scores are computed under the two-QTLs model as a LOD trace for one trait while keeping the QTL for the second trait fixed at its maximum-likelihood position, followed by subtracting the maximum LOD score of the single-QTL model in this region.[51] A maximum profile LOD value of zero (as for *CHS6*) indicates that the two-QTLs model fits no better than the single-QTL model. Positive maximum profile LOD values (as for *TRL1* and *IME2*) indicate that the two-QTLs model fits better than the single-QTL model, with the two profile LOD curves peaking at the best positions for the two QTLs. QTL and gene positions as shown in (A).

See also Figure S7 and Tables S3 and S4.

and growth. In contrast, the datasets analyzed here comprise multiple gQTLs for each condition and more than one eQTL for each of 5,495 genes. Most of these eQTLs are located on different chromosomes from their target genes and influence gene expression via *trans*-acting mechanisms. These multiple QTLs per trait or gene are caused by distinct, unlinked variants, providing geneti-

cally independent observations of the relationship between a gene's expression and a trait. To examine the relationship between eQTLs and gQTLs globally, we first divided the genome into 10-kb bins and counted the number of eQTL and gQTL peak markers in each bin. eQTL and gQTL counts were significantly correlated across bins (all eQTLs: Spearman rank

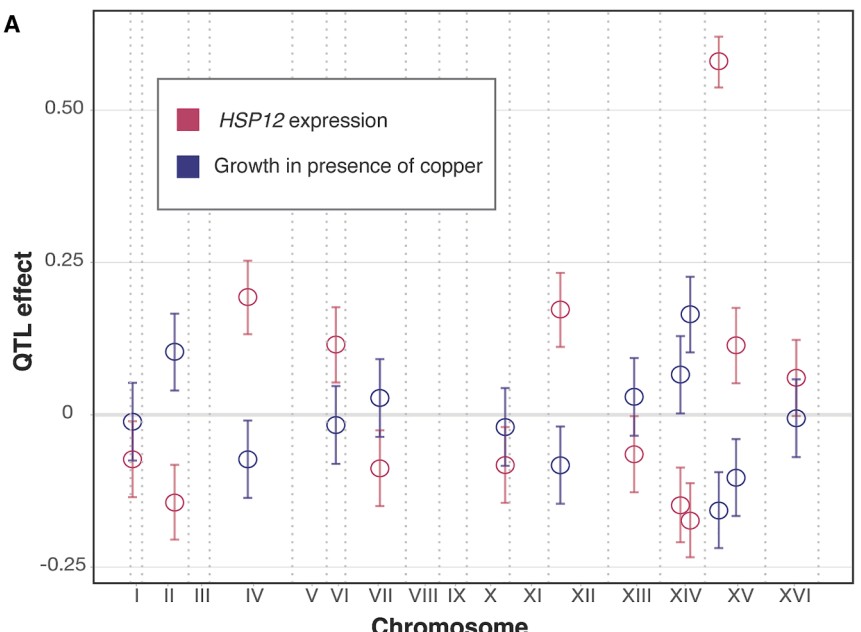

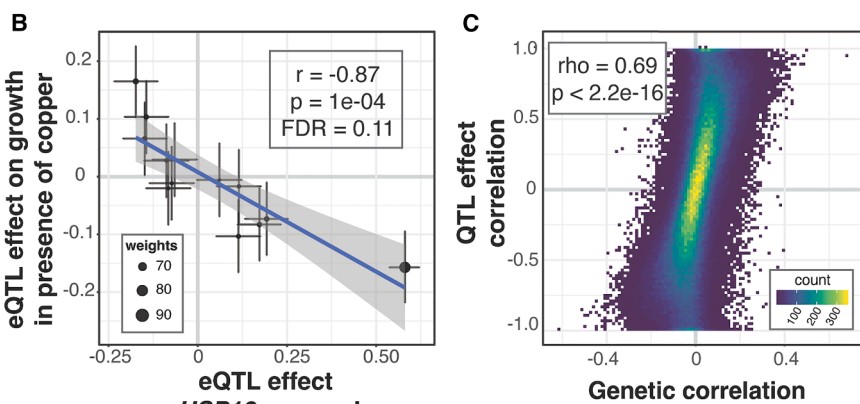

**Figure 3. QTL effect correlations**

(A) As an example, the plot shows the locations of eQTLs affecting the gene *HSP12* (*x* axis) and their effects (*y* axis; coefficient of correlation between trait and genotype) on *HSP12* expression (red) and growth in the presence of copper (blue). Shown here are 95% effect size confidence intervals. *HSP12* and copper were chosen as examples due to the large number of eQTLs and strong effect size correlation.

(B) Scatterplot of the effects from (A). The line represents the weighted regression. Weights are calculated as the inverse product of the widths of the 95% effect size confidence intervals and are reflected in the size of the point.

(C) Scatterplot between QTL effect correlation coefficients and genetic correlation coefficients across all genes and all growth conditions.

See also Tables S2, S3, and S5.

traits; up to 1,106 pairs for cycloheximide; Tables S2 and S3). These significant QTL effect correlations had very high coefficients (median absolute *r* = 0.92), as expected for correlations that are computed from few data points but nonetheless achieve an acceptable FDR.

If genetic correlations arise from multiple independent eQTLs across the genome, then we would expect correspondence between QTL effect correlations and genetic correlations. To test this, we examined the 28 traits that had at least 3 genes with significant QTL effect correlations. At 21 traits, genes with QTL effect correlations were enriched among genes with significant genetic correlations (FET uncorrected *p* < 0.05; Table S5). For 22 traits, there was a significant positive correlation between the magnitudes of the significant QTL effect correlations and their matched genetic correlations (Pearson correlation uncorrected *p* < 0.05; Table S5). Combining all traits and genes, there was strong correspondence between genetic correlations and the QTL effect correlations (ρ = 0.69, *p* < 2.2e−16; Figure 3C; this agreement was also seen when estimating the QTL effect correlations and the genetic correlations on randomly selected halves of the segregants: median ρ across 100 splits = 0.59).

In summary, we identified genes with multiple independent eQTLs that had consistent effects on the gene's expression and on growth in a given condition. These QTL effect correlations showed agreement with the genetic correlations, suggesting that the genetic correlations tend to arise from multiple eQTLs, most of which affect gene expression in *trans*.

### *Trans*-eQTL hotspots influence complex growth traits
The vast majority of *trans*-eQTLs in this yeast cross are clustered at 102 hotspot regions that each influence the

correlation ρ = 0.31, *p* = 9e−28; only *trans*-eQTLs: ρ = 0.30, *p* = 4e−27; similar results were obtained with 1- and 50-kb-sized bins: all ρ ≥0.16, all *p* ≤ 3e−16), indicating broad co-occurrence of loci that affect gene expression and growth.

If the precise abundance of a gene is important for a growth trait, then its independent eQTLs should affect the trait in a consistent fashion.[53] For example, if higher expression of a gene increases growth, then we would expect eQTLs that increase expression to also increase growth, while eQTLs that reduce expression would be expected to decrease growth. We searched for gene/trait pairs with such "QTL effect correlations." Briefly, we extracted the peak markers of all eQTLs for a given gene, computed their effects on gene expression and on growth, and performed weighted correlation tests on these effects (Figures 3A and 3B; STAR Methods). We examined 5,186 genes with at least 3 eQTLs and chose a lenient FDR of 20% (estimated within trait) because these analyses were based on a median of 6 loci, limiting statistical power. At this threshold, 2,038 gene/trait pairs showed a QTL effect correlation, distributed over 32 traits (median: 9 pairs among these

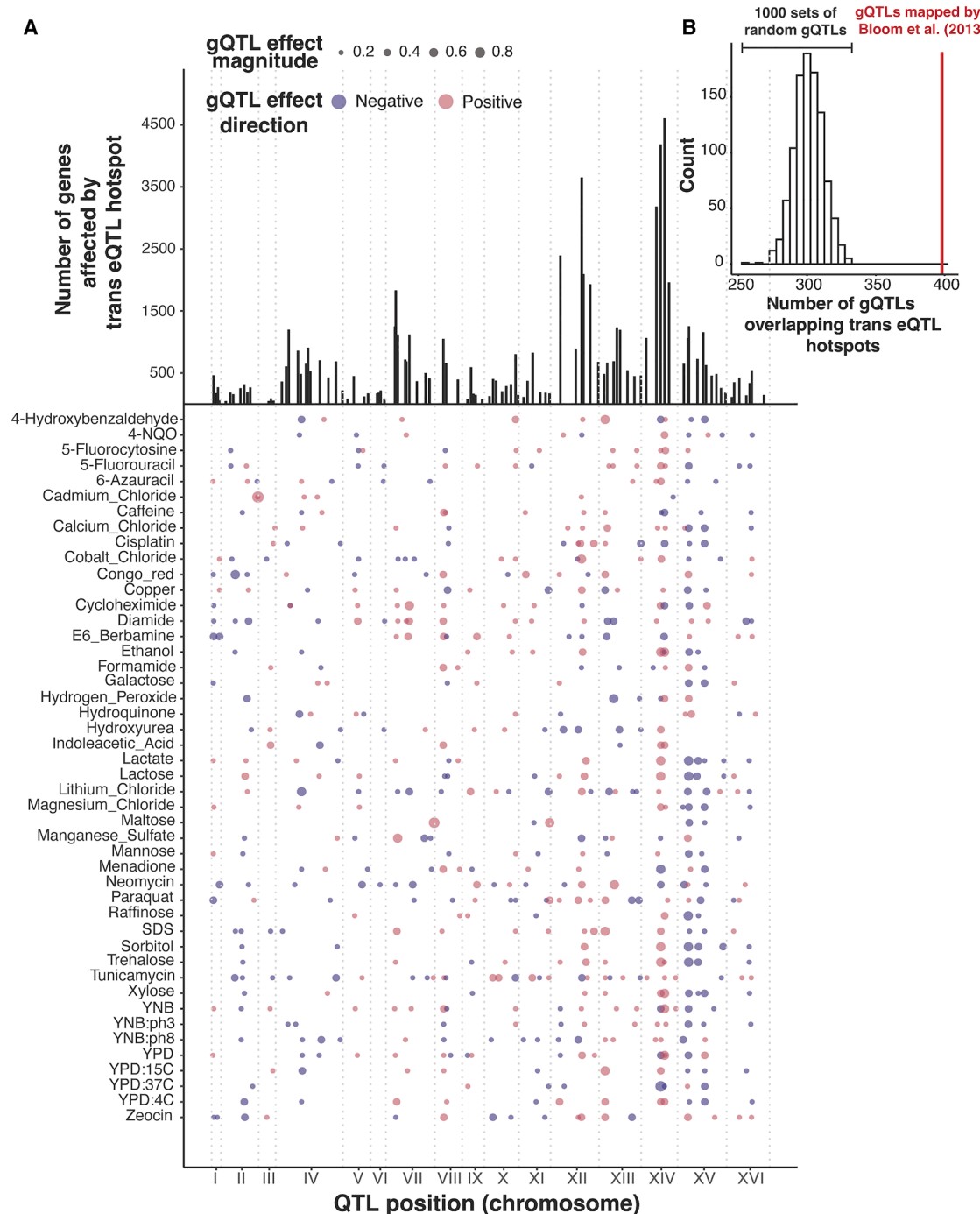

**Figure 4. *Trans*-eQTL hotspot and gQTL locations**

(A) Top: number of genes affected by the 102 *trans*-eQTL hotspots (vertical lines). Bottom: locations of the 591 gQTLs as points, with size scaled by effect size (coefficient of correlation between trait and genotype at the gQTL). Positive (RM allele increases growth compared to BY allele) and negative (RM allele decreases growth) gQTL effects are red and blue, respectively.

(B) Histogram showing the distribution of the number of gQTLs that overlap a *trans*-eQTL hotspot in 1,000 sets of randomly placed gQTLs, compared to the actual gQTLs (red line).

expression of dozens to thousands of genes (Figure 4A). Their broad effects suggest that *trans*-eQTL hotspots could also play a prominent role in shaping complex traits. Indeed, about two-thirds of gQTLs (398/591) overlapped at least one *trans*-eQTL hotspot (Figure 4A), exceeding random chance ($p < 0.001$; Figure 4B).

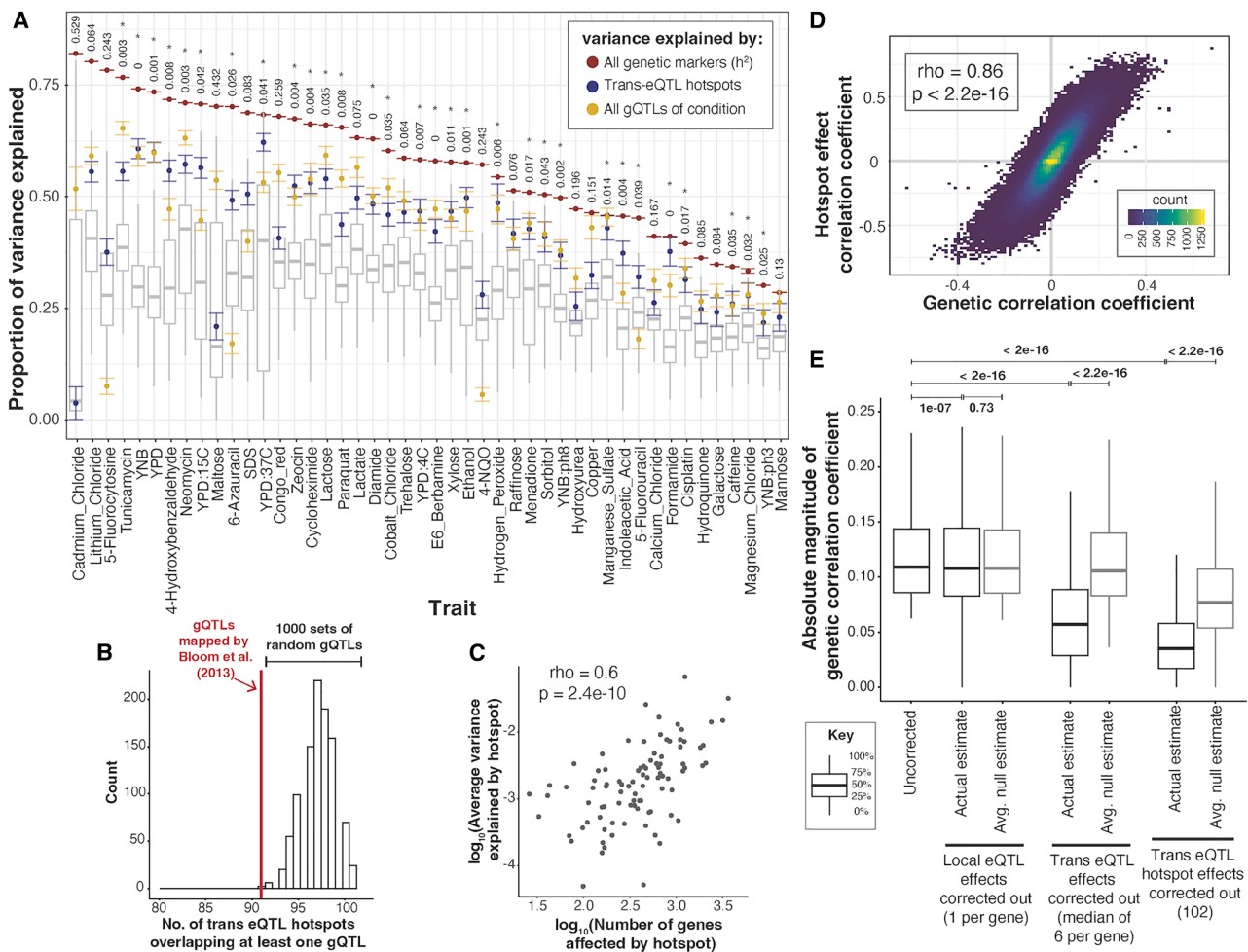

**Figure 5. Phenotypic variance and genetic correlations linked to *trans*-eQTL hotspots**

(A) Proportion of phenotypic variance explained by all 11,530 genetic markers (narrow-sense heritability; red dots), the 102 *trans*-eQTL hotspots (blue dots), and all gQTLs (yellow dots) in 46 growth conditions. Error bars about each estimate (depicted in the same color as the estimates) represent standard errors computed via 5,000 bootstraps. Boxplots show the distribution of variance explained by 1,000 sets of 102 random genetic markers. These distributions were used to calculate *p* values (shown at top) for the proportion of variance explained by the hotspots. *p < 0.05.

(B) Histogram showing the number of *trans*-eQTL hotspots that overlap at least one gQTL in 1,000 sets of randomly placed gQTLs, compared to the actual data (red line).

(C) Scatterplot comparing the average growth variance explained by each hotspot and the number of genes whose expression they influence. The top two hotspots influencing the expression of the most number of genes (chromosome [chr]XIV: 466588_T/G and chrXIV: 372376_G/A) are excluded from the analysis as it is not possible to estimate the average proportion of variance explained by just two markers under the linear mixed-model scheme we use to estimate the proportion of variance explained by hotspots.

(D) Scatterplot between hotspot effect correlation coefficients and genetic correlation coefficients across all genes and growth conditions.

(E) Boxplots showing distributions of magnitudes of genetic correlation coefficients before ("uncorrected") and after removing the effects of local eQTLs ("1 per gene"), *trans*-eQTLs (range 1–21 per gene, median of 6), and the 102 *trans*-eQTL hotspots (the number of markers regressed out for each category are indicated along the *x* axis). This analysis included only genes with at least one local eQTL and one *trans*-eQTL and only uncorrected genetic correlations that were significant at 5% FDR. For each category, a random expectation was computed by calculating the decrease in the magnitude of genetic correlations after regressing out the effects of 1,000 sets of random markers of size equal to the actual set. Wilcoxon *p* values for the difference in medians are indicated above the boxplots. See also Figures S8–S14 and Tables S2, S3, S5, and S6.

To quantify the influence of hotspots on growth variation, we fit mixed linear models to estimate the fraction of phenotypic variance in the 46 growth traits that is explained by DNA variation at *trans*-eQTL hotspots. We compared these values to the total additive heritability of the traits, estimated as the fraction of phenotypic variance due to all markers in the genome (Figure 5A).

Across the 46 conditions, a median of 77% of the heritability was captured by all 102 *trans*-eQTL hotspot markers (Table S2). To put these values in context, we computed the proportion of trait variance explained by the gQTLs of the given condition (STAR Methods). The amount of variance captured by the 102 hotspot markers corresponded to a median of 97% of the trait

variance explained by the gQTLs for the given condition (Figure 5A; Table S2). Given that the 102 hotspots are distributed widely across the genome and given the high degree of linkage in this one-generation cross, we asked whether the heritability captured by the hotspots could reflect random proximity to linked causal variants that are distinct from those that cause the hotspots. For 30 of the 46 conditions, the additive heritability explained by the hotspot markers was greater than that explained by 95% of 1,000 sets of 102 random genetic markers (Figure 5A; the random sets were in LD with similar fractions of the genome as the 102 real hotspots; Figure S8). On average, the random sets explained about 28.6% of the total additive heritability—much less than the actual hotspot positions. Thus, while some of the heritability assigned to hotspots probably arises from linked non-hotspot variants, *trans*-eQTL hotspots contribute a significant amount of heritability to yeast growth traits.

The hotspots vary widely in how many genes they affect (from 26 to 4,594), suggesting that their effects on growth are also likely to vary. Indeed, while nearly all (91 of 102) hotspots overlapped a gQTL, this number was slightly but significantly lower than expected (based on 1,000 sets of randomly placed gQTLs; $p < 0.001$; Figure 5B). Thus, while gQTLs are enriched at hotspots, not all hotspots are equally relevant. To explore this observation more quantitatively, we ranked the hotspots by the number of genes they affect, formed sets comprising an increasing number of hotspots (starting with the two hotspots affecting the most genes, and up to all 102 hotspots), and asked how much variance these sets explained in each of the 46 traits (Figure S9). Across traits, a median of 14 hotspots (with a range of 2–80) was needed to explain half of the maximum variance explained by all 102 hotspots, and a median of 41 hotspots (10–97) was needed to explain 80% of the maximum variance. Thus, while not every *trans*-eQTL hotspot in the baseline liquid YNB condition contributes to growth variation, dozens of hotspots do. Furthermore, the number of genes affected by a hotspot was positively correlated with the average proportion of variance that the hotspot explained across traits ($\rho = 0.6$, $p = 2.4\text{e}{-}10$; Figure 5C), suggesting that the average contribution of a hotspot to trait variation is related to the breadth of effects a given hotspot has on the transcriptome. To identify hotspots that are potentially causal for a specific trait, we quantified the fraction of the total variance explained by all hotspot markers in a given trait that is contributed by each hotspot as it is added to the analysis (Figure S10; Table S6). Causal hotspots should contribute a comparatively large amount of fractional variation. The results showed heterogeneity in which hotspots affect specific traits. Some hotspots contribute substantially to several traits (e.g., a hotspot at 541,139 bp on chromosome II, which affects growth on several carbon sources; a hotspot at 159,467 bp on chromosome I, which affects growth on several stressors), while other hotspots have more trait-specific contributions (e.g., a hotspot at 156,943 bp on chromosome IV that makes particularly strong contributions to growth on cycloheximide). No causal genes have been identified for these three hotspots, and their connections to diverse growth traits make them attractive targets for future dissection.

We asked whether the *trans*-eQTL hotspots are drivers of genetic correlations. To test this, we applied the procedure used to compute QTL effect correlations above to all *trans*-eQTL hotspot markers (STAR Methods). Briefly, we calculated the effects of the hotspot markers on the expression of each of 5,720 genes, as well as on growth in each of the 46 conditions. We then tested for correlation between these hotspot effects on gene expression and on growth (below, "hotspot correlations"; Figure S11).

Across the 46 conditions, a median of 1,704 genes had a significant hotspot correlation at an FDR of 5% (estimated within each trait), with a range of 4 genes (YNB at pH 8) to 3,795 (YPD) (Tables S2 and S3). Notably, these hotspot correlations agreed strongly with the respective genetic correlations for the same gene/trait pairs. This agreement was seen when combining all genes and all traits (Figure 5D), irrespective of gene expression level (Figure S12) and when estimating hotspot and genetic correlations in separate halves of the segregants (median $\rho$ across 100 random splits = 0.69). The agreement was also present for individual traits: 45/46 traits showed significant enrichment between genes with significant hotspot correlations and genes with significant genetic correlations (FET, uncorrected $p < 0.05$). For all 46 traits, the coefficients of the hotspot correlations and of the genetic correlations were strongly correlated across all genes ($p < 2.2\text{e}{-}16$, Table S5; median $r = 0.86$). For 32/46 traits, these observed correlations exceeded those seen in 95% of 1,000 random sets of 102 markers, showing that the high agreement between hotspot correlations and genetic correlations is not simply due to random linkage of hotspots to other causal variants. Overall, the strength of the agreement between hotspot correlations and genetic correlations suggests that *trans*-eQTL hotspots are major drivers of genetic correlations between gene expression and growth.

To directly contrast the contribution of different sets of loci to the genetic correlations, we tested how each genetic correlation changes when the effects of the given loci are removed. Specifically, for each gene/trait pair, we regressed out the gene's local eQTL, the gene's set of detected *trans*-eQTLs, or all *trans*-eQTL hotspots from both the gene expression and growth measurements. We then re-calculated the genetic correlations on the residual phenotypes. A drop in the strength of the genetic correlations would suggest that the removed locus was a contributor to the genetic correlation. Removal of each of the three QTL sets resulted in significant drops in the magnitude of the genetic correlations (Figure 5E). Similar patterns were seen for the individual growth traits, both for reductions in the strength and in the number of significant genetic correlations (Figures S13 and S14). Removal of *trans*-eQTLs and especially of the hotspots resulted in much larger reductions than removal of local eQTLs (Figure 5E; median drop in absolute genetic correlation coefficient: local eQTLs: 1%, *trans*-eQTLs: 48%, hotspots: 68%). *Trans*-eQTLs and hotspots but not the local eQTLs resulted in reductions that were larger than expected by chance (Figure 5E). These results highlight the importance of *trans*-acting eQTLs for complex trait variation.

## Major cellular states shape growth in specific conditions

To explore biological processes that connect gene expression and growth, we performed functional enrichment analyses. We focused on genetic correlations; enrichment results were broadly similar for QTL effect correlations and especially for

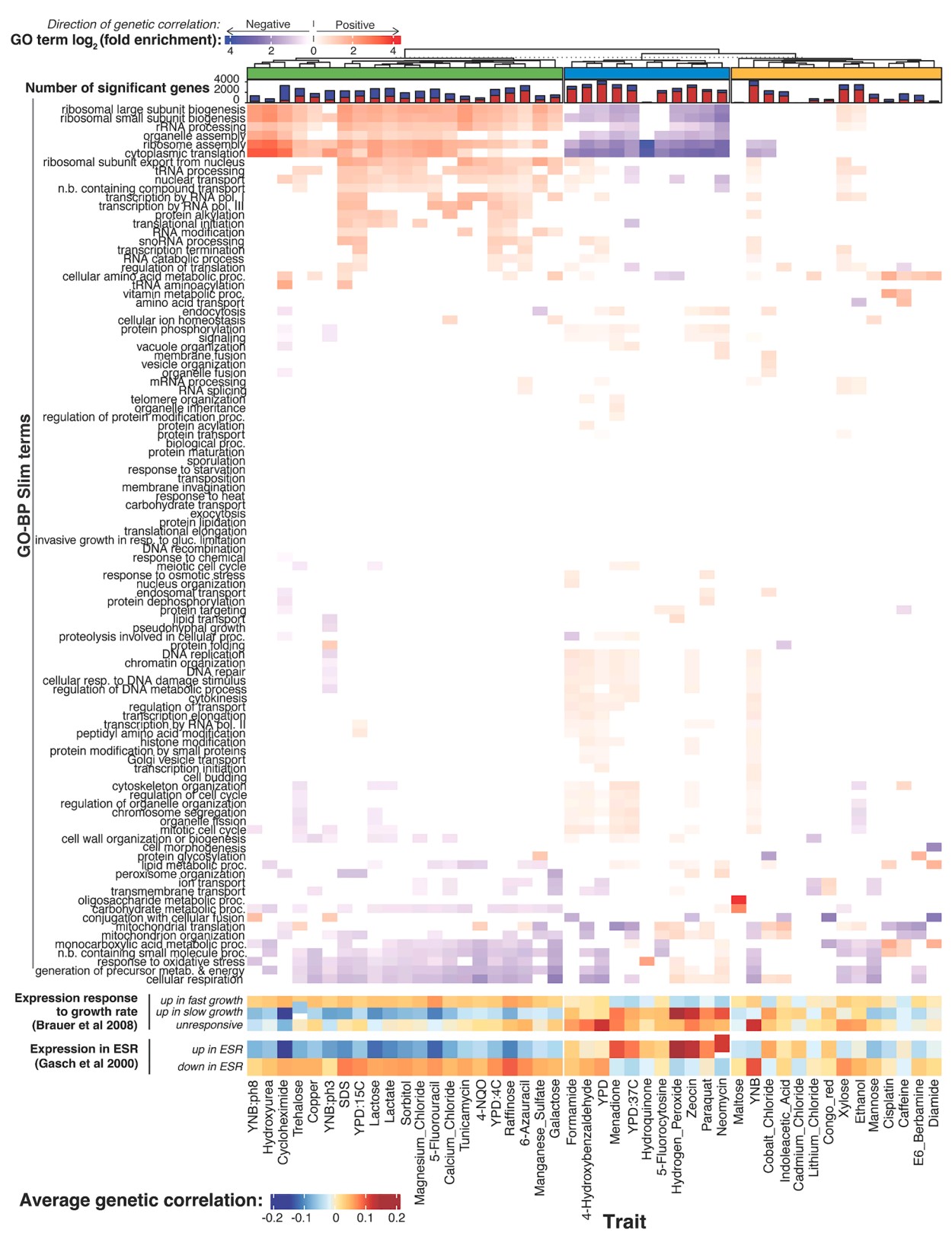

the hotspot correlations (Table S7). Genes with significant genetic correlations were enriched for a variety of biological processes as summarized in Gene Ontology (GO) slim annotations (Figure 6). Across all 46 traits, there were 503 enrichments ($p < 1e-5$; corresponding to Bonferroni correction for 100 GO categories tested in each of the 46 traits), with individual traits having between 0 and 27 enriched terms, with a median of 11.

Clustering traits based on their enrichment profiles revealed three distinct trait groups. For a group of 21 traits (shown in green in Figure 6), genes with positive genetic correlations were enriched for terms involved in ribosome biogenesis and translation, while genes with negative correlations were enriched for terms involved in respiration and carbohydrate metabolic processes. In yeast, there is a well-established relationship between gene expression and growth rate in chemostat cultures.[54] Genes with ribosomal functions are expressed more highly during faster growth, while genes involved in energy metabolism are expressed more highly during slower growth. We grouped genes based on published signatures of fast and slow growth[54] and found that the expression of genes in these signatures was correlated with growth in the green trait group (Figure 6). Furthermore, genes known to increase their expression during the environmental stress response[55] showed negative genetic correlations (Figure 6), while genes that decrease their expression during the stress response showed positive genetic correlations. Thus, higher growth in the conditions in the green trait group is associated with a genetic predisposition for higher translation and ribosome production, lower respiration, and a gene expression pattern that is the opposite of an acute stress response.

In contrast, a group of 11 traits that formed the blue trait group in Figure 6 tended to show enrichments for many of the same terms as the green trait group but in the opposite direction. Most notably, terms related to translation tended to be enriched among genes with negative rather than positive genetic correlations. The blue group was further divided depending on whether traits were positively or negatively correlated with mitochondrial GO categories. Some traits in this group (e.g., neomycin, hydrogen peroxide) showed strong positive correlations with the slow growth expression signature as well as with genes upregulated during the stress response. Thus, higher growth (as measured by larger colony sizes) in conditions in this trait group is associated with a genetic predisposition for lower translation, increased metabolism, and a gene expression pattern that resembles an acute environmental stress response. The remaining 14 traits in the orange group in Figure 6 showed more heterogeneous enrichments.

In summary, the genetic correlations revealed specific biological processes that are correlated with growth across several conditions. There is substantial heterogeneity in these relationships, such that growth in some conditions appears to be increased by a higher expression of genes involved in a certain process (e.g., translation, stress response), while growth in other conditions shows the opposite pattern.

### The *IRA2 trans*-eQTL hotspot mediates growth in hydrogen peroxide by modulating the expression of Msn2 target genes

Our genetic correlation analyses revealed that growth in different conditions is associated with specific biological processes. *Trans*-eQTL hotspots likely affect growth by altering the activity of these processes, as reflected by expression change at multiple genes. To explore this connection at a specific locus, we examined a hotspot caused by multiple missense variants in the *IRA2* gene.[56] This hotspot affects the expression of 1,240 genes[25] and overlaps gQTLs for 37 of the 46 conditions. Ira2 negatively regulates Ras-guanosine triphosphate signaling, resulting, among other consequences, in the expression of stress response genes by increasing active levels of the transcription factor Msn2. The *IRA2* RM allele is more active than the BY allele.[56,57] We therefore expected segregants carrying the RM allele to have a higher expression of stress response genes regulated by Msn2 than segregants carrying the BY allele and, therefore, to be better able to grow in stress conditions, including oxidative stress caused by hydrogen peroxide.[58] Indeed, the *IRA2* hotspot overlapped a gQTL for growth in hydrogen peroxide, where the RM allele increased growth compared to the BY allele ($r = 0.36$, $p = 3.5e-24$).

We asked which of the genes that are influenced by the *IRA2* hotspot mediate growth in hydrogen peroxide (Figure 7A). Mediation analyses revealed 380 genes at an FDR of ≤5% (Figure 7B; Table S8). These 380 genes were enriched for oxidoreductase function (hypergeometric test $p = 5e-7$), the oxidative stress response, and several biological processes associated with energy metabolism (Table S9). The 380 genes were enriched for regulatory targets of Msn2 (FET $p = 5.1e-14$; 291/380 genes; Table S8) and included established reporters of Msn2 activity (*HSP12*, *TPS2*, and *PNC1*).[59] The effect of the *IRA2* hotspot on growth in hydrogen peroxide was also mediated by its effect on the expression of *MSN2* itself, at an FDR of 12%. The proportion of the *IRA2* hotspot effect on growth in hydrogen peroxide that was mediated by the downstream targets of Msn2p (median: 10%, range: 2.5%–52%) was greater than that mediated by Msn2 itself (2.8%) (Figure 7; Table S8). These mediation analyses support the idea that the *IRA2* hotspot influences growth in hydrogen peroxide by influencing the expression of oxidative stress response genes regulated by Msn2. Ira2 may influence the expression of these genes primarily by altering the activity rather than the abundance of Msn2, which reacts to stress by

---

**Figure 6. Biological processes enriched in genetic correlations**

The top heatmap shows $\log_2$-fold enrichments for 100 GO-Biological Process slim terms in the set of genes with significant (5% FDR) genetic correlations. Enrichments for genes with positive and negative genetic correlations are shown in red and blue, respectively. Enrichments are displayed for terms significant at uncorrected $p < 0.001$. The stacked bar plot above the heatmap shows the number of genes with significant genetic correlation (5% FDR) for each condition, with positive genetic correlation in red and negative correlations in blue. The dendrogram clusters traits based on GO enrichment, with the three trait groups discussed in the text indicated. The heatmaps at the bottom show average genetic correlation coefficients for genes in the indicated gene groups from Brauer et al.[54] and Gasch et al.[55]

See also Table S7.

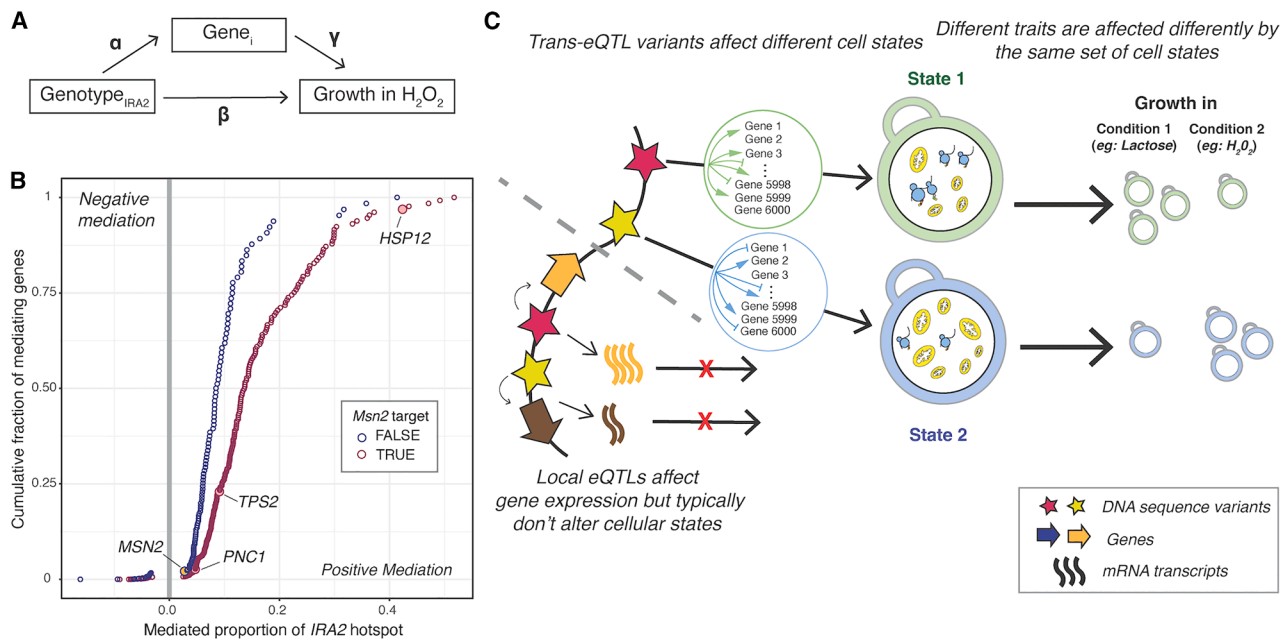

**Figure 7. Mediation of growth variation at a *trans*-eQTL hotspot**

(A) Conceptual model showing how the effect of the *IRA2* hotspot on growth in hydrogen peroxide may be mediated by the expression of hotspot target genes (*Gene$_i$*). The coefficients used in the mediation analysis (STAR Methods) are indicated.

(B) Mediated proportion of the *IRA2* hotspot effect on growth in hydrogen peroxide for 380 genes with significant mediation (5% FDR). Circles are colored according to their annotation as an Msn2 target in the Yeastract database. Cumulative fraction of mediating genes is computed for Msn2 targets and non-targets separately. Genes described in the text are highlighted. Positive mediation indicates genes for which increased expression increases growth, while negative mediation indicates genes for which increased expression decreases growth.

(C) Conceptual model of how *trans*-eQTL hotspots affect growth by altering cellular states, as reflected in the hotspot's effect on multiple genes. The two example states roughly correspond to the green and blue trait clusters in Figure 6, with high translation and low respiration versus low translation and high respiration. Growth in different conditions is affected differently by these states. Most local eQTLs (shown below the dashed gray line) affect gene expression but do not go on to affect key cellular states underlying growth variation.

See also Tables S8 and S9.

translocation to the nucleus upon phosphorylation.[60] The mediation effect on *MSN2* suggests that the precise abundance of Msn2 might also play a role.[61] This example illustrates how hotspots may orchestrate the expression of genes in a pathway to shape growth in a given condition.

## DISCUSSION

Our integrative analyses of eQTLs that alter mRNA abundance and gQTLs that shape yeast growth revealed key insights into the interplay of gene expression and complex traits. Most prominently, there were thousands of correlations between gene expression in a commonly used laboratory condition and growth in a variety of other conditions. Because gene expression was measured separately from growth, shared environmental factors are unlikely to have induced these correlations. Instead, the correlations arise from shared genetic effects on gene expression and growth. Similarly, cellular responses to the environmental conditions cannot have influenced our gene expression data. YNB medium is a possible exception, given that this medium was used to measure gene expression, but even here, growth and expression data were collected separately, with several

experimental differences. Therefore, our results support a model in which inherited alleles shape an individual's gene expression profile in a baseline condition in ways that can influence growth when cells encounter additional environmental factors not present in the baseline condition. In line with this model, earlier work performing gQTL and eQTL mapping in several environments showed that genes with eQTLs whose effects persisted across environments were more likely to be causal growth mediators than genes with eQTLs that were specific, and likely responsive, to a given growth condition.[46]

The contribution of local eQTLs to genetic correlations was statistically significant but modest in magnitude (Figure 5E). This result mirrors recent work in humans showing that *cis*-eQTLs tend to be underrepresented at genes identified as GWAS hits for complex traits, in part because negative selection reduces the frequency of variants with strong effects on the expression of genes that are most relevant for complex traits, making them hard to detect with limited sample sizes.[24] While nearly half of genes in the BY/RM cross are affected by a local eQTL,[25] negative selection may still have reduced the incidence of local eQTLs at the most trait-relevant genes. Even though there is a growing list of local eQTLs that have been shown to

underlie QTLs for complex traits (for examples in yeast, see Fay et al.[62] and Salinas et al.[20]), the large number of local eQTLs in this cross restricted our ability to systematically annotate gQTLs to specific causal local eQTLs. gQTLs can also be caused by coding variants (for examples in yeast, see Fay et al.[62] and Sadhu et al.[39]) that may not act via altered expression of the causal gene and do not involve any of the local eQTLs in the given region.

In contrast, the joint effects of multiple *trans*-eQTLs per gene were a major source of genetic correlations between gene expression and growth (Figure 5E). Enabled by the large number of mapped *trans*-eQTLs in this cross, we used these loci as genetically independent observations to search for consistent relationships between expression change at a given gene and growth in a given condition. This strategy is similar to analyses of allelic series at a given gene in human populations, in which alleles of varying molecular severity, ranging from knockouts to subtle *cis*-eQTLs, are assayed for their effects on a complex trait to create dose-response curves.[63,64] Related genetic strategies and synthetic approaches have been used to relate the expression level of several focal genes to yeast fitness[65–68] and human gene regulation.[69,70] Here, we applied this logic to independent, mostly *trans*-acting eQTLs across the genome and to dozens of traits.

More than 90% of *trans*-eQTLs in this cross occur at hotspot positions. By applying effect size correlations to *trans*-eQTL hotspots, we found that the joint effects of the hotspots contribute strongly to growth trait heritability and genetic correlations between gene expression and growth. Together, our results showed that genetic correlations between gene expression and growth are typically not caused by single eQTLs with strong effect (e.g., local eQTLs) but instead by the joint action of a collection of loci across the genome, most of which affect mediating genes in *trans*.

Enrichment analyses of the genetic correlations suggested certain functional gene groups as important mediators of complex trait variation. The same GO terms tended to be enriched across traits (Figure 6), highlighting central biological processes with broad importance for many traits. However, there were differences in the direction of these enrichments.

We summarize these observations in a model centered on key cellular processes such as translation, biomass production, and metabolism (Figure 7C). These processes are in a balance, akin to a recently described cell state axis related to protein kinase A and target of rapamycin signaling, along which genetically different strains vary quantitatively.[71] The activity of these key processes and the state of the cell is reflected in coordinated abundance variation of hundreds of genes. Complex growth traits are shaped by variation in the key processes as a whole, rather than by the precise abundance of most individual genes that constitute a given process. Depending on specific trait biology, growth in some conditions is increased by a cell state that is genetically biased toward high biomass production and fast growth, along with a dampened stress response. This pattern is seen in the green trait group in Figure 6, which includes several alternative carbon sources. Growth in other conditions may be increased by a genetic predisposition for a stronger stress response. Examples are traits in the blue trait group in

Figure 6, which includes stressors such as high temperatures and oxidative stress caused by $H_2O_2$. In these conditions, genetic predisposition for an increased baseline stress response signature may play a role similar to that of the phenomenon of inducible adaptive stress responses, in which exposure to low levels of one stressor protects cells from subsequent exposure to a different stressor.[72–76] Previous work has revealed examples where the expression of a given gene can be beneficial or detrimental for growth depending on the environment.[46] Future work will need to disentangle exactly how genetic effects on the specific processes found here shape growth.

Combining the above model with our QTL observations suggests a two-tiered architecture for how eQTLs shape complex traits (Figure 7C). Despite their prevalence and stronger effects on gene expression,[25] many local eQTLs appear to have little influence on growth traits because they do not alter the relevant central processes.[21,24,71] By contrast, some DNA variants do rise above this molecular background of functionally inert local variation and do affect cellular state and growth. These variants can affect cell states directly or indirectly, can be coding or noncoding, and may occur at a diverse set of genes throughout the genome.[77] When a variant does affect cell state, this effect becomes detectable as expression changes at the many genes whose abundance constitutes and reflects the state, resulting in the observation of a *trans*-eQTL hotspot. Hotspot effects on critical cell states explain the overall enrichment of *trans*-eQTL hotspots at gQTLs we observed here (Figure 4).

### Limitations of the study

Our study has several limitations. Gene expression in a single baseline condition provides a good model for reference eQTL panels commonly used in other species[12,37,44,45,78] and escapes issues with reverse causation, but it cannot reveal trait causation by regulatory genetic effects present only in a given condition.[21,22,57,79–83] Linkage among neighboring variants remains a key barrier for resolving loci and causal relationships. Power remains limited even in the genetically comprehensive datasets studied here. That most detected QTLs have small effects[25,26] makes it likely that large numbers of undetected eQTLs and gQTLs remain, each of very small effect. Imperfect power and linkage combine to produce wide QTL confidence intervals, which are difficult to use in colocalization analyses. Larger segregant panels are becoming available[84] but still suffer from linkage. Methods to directly engineer single variants in parallel[39,85–89] could be deployed in large samples and diverse conditions to ultimately achieve dissection of the complex interplay between gene expression and growth variation at gene-level and variant-level resolution.

### Conclusion

In summary, we revealed a prominent role of *trans*-eQTL hotspots in shaping genetically complex traits. The *trans*-eQTL hotspots were detected in a baseline condition that was different from the tested growth conditions, and they likely arise from genetic variants that alter critical cell states. These states serve as points of convergence for diverse genetic variants across the genome. When the cell is confronted with a non-baseline environment, this altered cell state provides a genetic predisposition

**Cell**Press

that can help or hinder growth, depending on the specific biology of the trait.

## RESOURCE AVAILABILITY

### Lead contact

Further information and requests for resources should be directed to and will be fulfilled by the lead contact, Frank W. Albert (falbert@umn.edu).

### Materials availability

This study did not generate new unique reagents.

### Data and code availability

All analysis code is available at https://doi.org/10.5281/zenodo.15122168. This study did not generate new datasets. Colony sizes for individual segregants were obtained at http://genomics-pubs.princeton.edu/YeastCross_BYxRM/ from the file BYxRM_PhenoData.txt. gQTLs analyzed in the present work were obtained from table S3 in Bloom et al.[26] Expression values were obtained from Source Data 1 at https://doi.org/10.7554/eLife.35471.021. eQTLs were obtained from Source Data 4 at https://doi.org/10.7554/eLife.35471.024.

## ACKNOWLEDGMENTS

We thank Joshua Bloom, Randi Avery, Sheila Lutz, Kelsey Johnson, Mahlon Collins, and Emma Sol Taylor-Brill for discussions and comments on the manuscript. We thank Qiuming Wu for preliminary downsampling analyses. This work was funded by NIH grant no. R35GM124676 awarded to F.W.A.

## AUTHOR CONTRIBUTIONS

Conception, K.R. and F.W.A.; formal analysis and investigation, K.R.; funding acquisition and supervision, F.W.A.; writing – original draft and writing – review & editing, K.R. and F.W.A.

## DECLARATION OF INTERESTS

The authors declare no competing interests.

## STAR★METHODS

Detailed methods are provided in the online version of this paper and include the following:

- KEY RESOURCES TABLE
- EXPERIMENTAL MODEL AND STUDY PARTICIPANT DETAILS
- METHOD DETAILS
- QUANTIFICATION AND STATISTICAL ANALYSIS
  - Data preparation
  - Genetic correlations
  - Effect of segregant panel size
  - Colocalization tests
  - QTL effect correlations
  - Overlapping *trans*-eQTL hotspots and gQTLs
  - Heritability estimates
  - Hotspot effect correlations
  - eQTL sources of genetic correlations
  - GO enrichment analyses and trait clustering
  - Comparison with published datasets
  - Mediation

## SUPPLEMENTAL INFORMATION

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

**CellPress**

# STAR★METHODS

## KEY RESOURCES TABLE

| REAGENT or RESOURCE | SOURCE | IDENTIFIER |
| --- | --- | --- |
| **Deposited data** | | |
| Gene expression data | Albert et al.[25] | https://doi.org/10.7554/eLife.35471.021 |
| eQTL data | Albert et al.[25] | https://doi.org/10.7554/eLife.35471.024 |
| Yeast colony size data | Bloom et al.[26] | http://genomics-pubs.princeton.edu/YeastCross_BYxRM/ |
| Growth QTL data | Bloom et al.[26] | Table S3; https://www.nature.com/articles/nature11867 |
| **Software and algorithms** | | |
| Analysis code for this study | This paper | https://doi.org/10.5281/zenodo.15122168 |
| R programming language | The R Project for Statistical Computing | https://www.r-project.org/ |

## EXPERIMENTAL MODEL AND STUDY PARTICIPANT DETAILS

The eQTL and gQTL data analyzed in this study were gathered in a panel of 1,012 *S.cerevisiae* meiotic segregants generated by crossing the ''BY'' prototrophic laboratory strain (a close relative of the genome reference strain S288C; MAT**a**) and the ''RM'' vineyard strain (*MATα hoΔ::hphMX4 flo8Δ::natMX4 AMN1-BY*). Each of the segregants had been genotyped at 42,052 markers. Markers in perfect linkage had been collapsed into a set of 11,530 unique markers,[25] which we used for the analyses in this paper unless otherwise specified.

## METHOD DETAILS

All data analyzed in this paper was previously published. Bloom et al. measured growth of the segregants in 46 conditions.[26] Segregants were pinned to agar plates containing 50 mL of a solid base medium (YNB or YPD) with the given drug or condition (Table S1), at a density of 384 segregants per plate. After 48 h of growth, colony size was measured by plate imaging. We used these endpoint colony sizes (from the file ''BYxRM_PhenoData.txt'' at http://genomics-pubs.princeton.edu/YeastCross_BYxRM/) for individual segregants as relative measures of yeast growth. We sometimes refer to these growth measurements simply as ''traits''. Using these traits and the segregant genotypes, Bloom et al. mapped a total of 591 gQTLs across 46 conditions.[26] gQTLs analyzed in the present work were obtained from Table S3 in Bloom et al.[26]

Albert, Bloom et al.[25] grew each of the segregants in liquid YNB medium in 96-well plates, with a different arrangement of segregants into plates than in Bloom et al.[26] RNA was extracted at an average optical density at 600 nm ($OD_{600}$) of ∼0.4. Subsequently, they performed RNA sequencing to quantify the expression of 5,720 genes as the log2(transcripts per million +0.5). We used these expression values (obtained from Source Data 1 in Albert et al.[25]: https://doi.org/10.7554/eLife.35471.021) as the gene expression measurements in the current paper. Principal component (PC) analyses showed that these transcriptomes are diverse, with 63 PCs needed to account for 80% of variation (Figure S15). The first 10 PCs all showed significant GO enrichments (Figure S16), suggesting that these PCs reflect biological variation.

Albert et al.[25] used these data to map 36,498 eQTLs across 5,643 genes at 5% FDR. An eQTL was defined as local if its confidence interval overlapped the position of the gene it affected, where the gene position was considered to comprise the coding sequence plus 1,000 bases of upstream and 200 bases of downstream sequence. Using these criteria, 2,884 genes had a local eQTL. The number of eQTLs across the 5,643 genes ranged from 1 to 22, with a median of 6 eQTLs per gene. Except for a single local eQTL per gene (where present), these eQTLs all act in *trans*, comprising the vast majority of the dataset. eQTLs analyzed in the present work were obtained from Source Data 3 in Albert et al.[25]

## QUANTIFICATION AND STATISTICAL ANALYSIS

All analysis code is available at https://doi.org/10.5281/zenodo.15122168. All analyses were conducted in R version 4.3.1 or R version 3.5.2 (https://www.r-project.org/).

## Data preparation

We analyzed data from 979 segregants that had both gene expression measurements and growth measurements in at least one condition. We corrected gene expression for the batch (i.e., the 96-well plate a segregant was grown in) and

growth covariates (OD$_{600}$ at the time of RNA extraction) reported in Albert et al.,[25] using the following linear model for each gene:

$$E \ = \ \alpha B + \beta G + R_{expression}$$

Here, $E$ is a vector of uncorrected gene expression measurements across the segregants. $B$ and $G$ are vectors of segregant batch IDs (modeled as a categorical factor) and OD$_{600}$ measures (modeled as a numerical covariate), respectively. $\alpha$ and $\beta$ are the estimated fixed-effect coefficients for the batch and growth covariates, respectively. $R_{expression}$ is the vector of gene expression residuals after correction for batch and growth covariates. We used $R_{expression}$ in all downstream analyses. No corrections were applied to growth measurements. We standardized the growth and corrected gene expression measures to have a mean of 0 and standard deviation of 1, using the 'preProcess' function in the *caret* R package.[90]

### Genetic correlations

We computed the genetic correlation between the expression of a given gene and growth in a given condition as the Pearson correlation coefficient across segregants. While correlations between phenotypes (such as, in our case, gene expression and colony size) can in principle be caused by shared environmental factors in addition to shared alleles, the numerous experimental differences between the two studies analyzed here (see above) make environmental sources of these correlations extremely unlikely. We therefore consider the correlations computed here as genetic in origin. We estimated the FDR using the $q$ value package,[96,97] estimated within each trait.

The growth traits analyzed here are not entirely independent from each other, as reflected by significant correlations among them (Figure S1; see also Bloom et al; [26]). This raises the possibility that the genetic correlations observed for the 46 traits may reflect the same underlying biological relationships, rather than trait-specific connections. To test this, we explored whether the genetic correlations for different traits were driven by factors that are shared among the 46 traits.

#### *Correction for growth on YNB or YPD base medium*

Growth traits were measured as colony sizes on solid agar plates with either YNB or YPD as base media, which were augmented with various treatments (Table S1). This means that genetic influences on how well a segregant grows on these two base media could also influence growth in all conditions that use the respective base medium. Such shared effects of growth on YNB or YPD could be reflected in expression-trait correlations that recur across traits.

To test for the influence of the solid agar medium, we removed the effects of the base medium $b$ on growth in a given condition $i$ using the following formula:

$$G_i \ = \ \alpha G_b + R_{growth,i}$$

Here, $G_i$ is the vector of growth in condition $i$. $G_b$ is the vector of growth in the given base medium $b$, where $b$ can be either YNB or YPD, depending on which base medium the given condition had been added to. $\alpha$ is the estimated coefficient for the effect of the base medium on growth in the given condition. $R_{growth,i}$ is the vector of residuals, containing the growth in the given condition after correcting for the growth in the base medium. Subsequently, we re-computed the genetic correlation between the expression of each gene with the vector $R_{growth,i}$. Results are shown in Figure S2.

#### *Shared factors across growth traits*

To explore whether unknown shared factors beyond the solid medium could account for covariation across conditions, we performed a principal component analysis on the matrix of growth traits. We used the 42/46 growth traits with measurements for at least 20% of total segregants; the excluded growth conditions were cadmium chloride, hydrogen peroxide, raffinose, and sorbitol (Table S1). This analysis considered 457/979 segregants with growth measurements for all of these 42 conditions. Any strong shared factors shaping growth across conditions would be reflected in large amounts of variation attributed to the first few principal components. Instead, the first principal component accounted for just 16.5% of variance among 42 traits. Cumulatively, the first five principal components explained only 42% of variance (Figure S3). Together, these analyses show that there are no strong common factors that drive most of the variation among traits, and that most of the correlations between gene expression and growth traits are specific to each trait.

### Effect of segregant panel size

To investigate how the size of the segregant panel affects the number and magnitude of detected significant genetic correlations, we randomly subsampled without replacement from the total 979 segregants to create subsets of 250, 500, and 750 segregants, with 5 random subsets for each sample size. For each subsampled segregant panel, we computed genetic correlations as above. For each subset size, we recorded the median number and magnitude of significant (at 5% FDR) genetic correlations for each of the 46 conditions across the five random subsets.

### Colocalization tests

We performed colocalization tests for experimental crosses as implemented in Boehm et al.[51] Their test conducts two-dimensional scans for models with a single pleiotropic QTL or two distinct QTLs in a given interval known to contain QTLs for two traits. Here, these two traits were growth in a given condition and the expression of a given gene that is physically located in the same region (i.e., we

analyzed gQTLs that overlapped local eQTLs). The test identifies the QTL positions that maximize the likelihoods under the assumption that the two traits are influenced by (a) a single pleiotropic QTL that arises from shared causal variants, or (b) two distinct QTLs arising from different causal variants. The log-ratio of the likelihoods of these two models is the test statistic, whose statistical significance is determined by parametric bootstrap. A significant $p$-value in this test rejects the null hypothesis of one pleiotropic QTL in favor of two distinct QTLs for the two traits. Therefore, a $p$-value greater than 0.05 implies pleiotropy in that a model of shared variants could not be rejected. We ran these colocalization tests using the package $qtl2pleio$.[51]

We applied the test to instances of local eQTLs overlapping a gQTL. QTL overlap was judged based on the genomic confidence intervals of the QTLs, extended by 5 kb on each side. The colocalization test is better able to detect distinct QTLs when the QTLs have larger effects. Therefore, we focused our analyses on 2,074 cases where the overlapping local eQTLs and gQTLs both have a LOD score $\geq$10, to increase the chance that a failure to reject the null hypothesis of pleiotropy indeed suggests pleiotropy rather than an underpowered test. These 2,074 pairs comprised 188 gQTLs from 45 conditions that overlapped with at least one local eQTL for one of 581 genes. The pleiotropic model was not rejected at about half (1,052) of the QTL pairs. At these QTL pairs, which included 95% (178/188) of the analyzed gQTLs, the same DNA variants may cause both the local eQTL and the gQTL, and we analyzed these 1,052 pairs further.

Boehm et al. 's test has high power to detect two traits as having distinct QTLs when their peaks are separated by more than 1 centiMorgan (cM), which in our data corresponds to a correlation of neighboring marker genotypes of r < 0.98. To preserve power while reducing run time, we performed our colocalization tests with a reduced set of 1,496 markers for which consecutive marker genotypes had r < 0.95. We expected this reduced marker set to decrease the power to detect distinct QTLs, perhaps increasing the number of "false pleiotropy" assignments. To examine the severity of this bias, we re-tested an independent set of 122 QTLs that had all been called to be pleiotropic based on the reduced marker set (r < 0.95) with a denser set of 3,025 markers with r < 0.98. Even with this denser set of more highly correlated markers, pleiotropy was rejected at only 5 of these 122 QTLs (∼4%). Given the majority of pleiotropy calls were thus the same when using the set of markers with r < 0.95 and those with r < 0.98, we chose to run our colocalization tests with the set of genomic markers at r < 0.95 to save computational time.

In the colocalization tests, we considered the smallest genomic interval that contained the confidence intervals for both the local eQTL and the gQTL, padded by 5 kb. For $p$-value estimation, we ran 1,000 parametric bootstraps. We considered two QTLs to be pleiotropic when the null hypothesis of pleiotropy was not rejected ($p$ > 0.05) and when there were at least three genomic markers in the examined interval. We did not perform multiple-test correction for these analyses because our goal was to identify QTLs that are pleiotropic (i.e., those for which the test is not significant). A liberal criterion for rejecting pleiotropy (without correcting for multiple testing) is conservative for this purpose.

### QTL effect correlations

We estimated the magnitudes of genetic effects on expression and growth at eQTLs detected in Albert et al.[25] as follows. For each gene, we extracted the peak markers of its detected eQTLs, as reported in Source data 4 (https://doi.org/10.7554/eLife.35471.024) from Albert et al.[25] At each of these peak markers, we obtained QTL effects on gene expression by computing the Pearson correlation coefficient between the expression of the gene in the segregants and their genotypes at the peak marker. Similarly, at the same markers, we obtained QTL effects on growth by computing Pearson correlation coefficients with growth in each of the 46 conditions.

We then computed the weighted Pearson correlation between these QTL effects on gene expression and QTL effects on growth, for each of the 46 conditions as follows. For computing the weights, we first calculated the widths of the 95% confidence intervals around the expression and growth effects using the $cor.test$ function in R and then took the inverse of their product. We subsequently passed these weights to the $wtd.cor$ function in the $weights$ R package (https://CRAN.R-project.org/package=weights) to get the weighted Pearson correlation coefficients between the QTL effects on expression and growth. We estimated the FDR within each trait using the "fdr" method in the $p.adjust$ function, which implements the Benjamini & Hochberg correction.[91]

### Overlapping $trans$-eQTL hotspots and gQTLs

We counted the number of overlaps between the confidence intervals of the gQTLs for a given condition with the confidence intervals of the 102 $trans$-eQTL hotspots identified in Albert et al.,[25] as reported in their Source Data 8 (https://doi.org/10.7554/eLife.35471.028). For each condition, we computed the random expectation of these values as follows.

For a condition with $N$ gQTLs, we selected $N$ random non-overlapping genomic intervals of the same sizes as the actual gQTLs. Subsequently, we counted overlaps between these random gQTLs with the observed (i.e., non-randomized) hotspots. We repeated this 1,000 times per condition to obtain a trait-specific distribution of the expected number of gQTLs that overlap at least one $trans$-eQTL hotspot by chance. Using this distribution, we calculated $p$-values as the fraction of random sets that matched or exceeded the observed number of gQTL/hotspot overlaps. We considered an uncorrected $p$ < 0.05 to be significant in this analysis.

### Heritability estimates

For each of the 46 conditions, we computed the proportion of growth variance explained by a given set of genomic markers using the following linear mixed model, fit using the $lme4qtl$ package[92]:

$$Growth_i \sim (1 \mid G_n)$$

Here, '$Growth_i$' is the growth of 979 segregants in condition $i$, and $G_n$ is the genetic relatedness matrix generated from the given set of $n$ genomic markers. We then estimated the fraction of growth variance explained by $G_n$. We implemented this model for the following sets of genomic markers: (a) all 11,530 genetic markers with between-genotype correlations less than 1; (b) 102 markers corresponding to the peak markers of the *trans*-eQTL hotspots from Albert et al.[25]; (c) for each condition, the markers corresponding to the peak markers of their respective gQTLs. We estimated the standard errors of heritability estimates by performing a bootstrap resampling procedure with 5,000 iterations.

This one-generation cross is subject to high degrees of linkage among neighboring variants, such that the heritability estimated from a given marker can capture contributions from causal variants in the vicinity. To control for this, we estimated the heritability captured by sets of 102 random genomic markers by fitting the above model at 1,000 sets of 102 random markers from the total set of 11,530 markers, for each of the 46 conditions.

For the analyses in Figure S9, we ordered the 102 hotspots by the number of genes they affect, from many to few affected genes. We then fit the mixed linear models above on genetic relatedness matrices estimated from hotspots of increasing size, starting with the two hotspots that affected most genes, and up to all 102 hotspots.

### Hotspot effect correlations

We computed the effects of the 102 *trans*-eQTL hotspots from Albert et al.[25] on expression of each of 5,720 genes and growth in 46 conditions from Bloom et al.[26] using the following stepwise approach.

For each gene or growth trait, we computed the correlation coefficients between gene expression or growth and the genotype at each of the 102 hotspot peak markers. We retained the strongest significant correlation among the set with a correlation $p$-value $<0.05$, and residualized the gene expression or growth measures to remove the effect of this strongest marker. We then used the residualized gene expression or growth values to calculate the correlation coefficients with each of the remaining 101 markers. We repeated this procedure until there were no more markers with correlation $p < 0.05$, and set the remaining effect estimates to zero. Because the hotspot locations had been previously identified using a rigorous detection process Albert et al.,[25] our procedure here did not attempt to correct for false discoveries. Instead, our goal was to assign effect estimates to any hotspot markers with a non-zero effect, based on a relaxed significance criterion.

Subsequently, we computed the hotspot effect correlation for a given gene/trait pair as the Pearson correlation coefficient between the effects of the hotspot markers on the expression of the gene and on the growth in the given growth condition. We estimated the FDR using the "fdr" method in the *p.adjust* function, separately for each trait.

To estimate the distribution of hotspot effect correlations expected by chance, we used 1,000 sets of 102 random genetic markers as in the 'heritability estimates' section and repeated the procedure above.

### eQTL sources of genetic correlations

To quantify the contribution of different sets of eQTLs to the genetic correlations, we regressed out the effects of the respective loci from the expression and growth data using the following linear models.

$$G_i = \sum_{k=1}^{n} \alpha_k * Genotype_k + R_{growth,i}$$

$$E_j = \sum_{k=1}^{n} \beta_k * Genotype_k + R_{expression,j}$$

$G_i$ and $E_j$ are the vectors of standardized growth measurements in condition $i$ and the standardized expression measurements for gene $j$, respectively. *Genotype* represents the vectors of genotypes at $n$ loci. $n$ is either the number of local eQTLs for gene $j$ (usually here $n = 1$), the number of *trans*-eQTLs for gene $j$, or $n = 102$ for the *trans*-eQTL hotspots. $\alpha$ and $\beta$ are the estimated fixed effect coefficients for the corresponding genotypes. $R_{growth,i}$ and $R_{expression,j}$ are the vectors of the residuals, containing the growth and expression values for the given condition and gene respectively after regressing out the effects of the given marker set.

We then used $R_{growth,i}$ and $R_{expression,j}$ to re-compute the genetic correlation coefficients. Genetic correlation coefficients for all 5,643 genes and 46 conditions before and after regressing out locus effects were compared using paired Wilcoxon tests.

### GO enrichment analyses and trait clustering

For each of the 46 conditions, we performed GO enrichment analysis on the genes with significant genetic correlations at 5% FDR using the following approach. First, we divided the genes with significant genetic correlations into two groups based on the directions of the correlations ('*positive*': r > 0, '*negative*': r < 0). We then performed Fisher's exact tests on these gene lists to test for significant enrichment for 100 GO Slim-biological process (BP) terms.[93] We used either the "positive" or the "negative" enrichment in downstream analyses, based on the significance of the GO-term at $p < 0.001$. When a GO-term was significant in both the "positive" and "negative" enrichment analyses, we retained the GO-term in the analysis with the greater magnitude of log-fold enrichment.

We clustered the different growth conditions into trait groups using the 100 GO-BP terms. For each condition, we calculated the fold-enrichment of each of the 100 GO-BP terms as the ratio of the number of genes in our gene list annotated to the given GO term

and the number of genes expected to be observed for this GO term. We clustered the traits based on fold-enrichments for GO-BP terms that were significant at nominal $p < 0.001$. We performed k-means clustering using the *ComplexHeatmap* package[94] and chose to cluster the traits into 3 groups based on visual inspection of the patterns of enrichments across all GO term/trait pairs (Figure 6).

### Comparison with published datasets

We compared our genetic correlations with results from two prior gene expression studies. First, Brauer et al. studied the relationship between gene expression and growth rate in chemostat cultures.[54] They identified three groups of genes based on their expression response to different growth rates: genes whose expression is upregulated in fast growth ($n = 258$), genes upregulated in slow growth ($n = 367$), and genes that are unresponsive to growth rate ($n = 980$). For each trait, we computed the average genetic correlation coefficients in our analyses for the genes belonging to these three groups and plotted them along with the GO term fold enrichment signatures for all 46 growth traits. We similarly computed the average genetic correlation coefficients for the groups of genes that were found in Gasch et al.[55] to be upregulated ($n = 269$) and downregulated ($n = 570$) during the environmental stress response, and plotted them along with the GO term fold enrichment signatures.

### Mediation

We performed mediation analyses using the *mediation* package[95] for each of the 1,240 genes regulated by *IRA2* hotspot (as reported in Source Data 9 in Albert et al.[25]: https://doi.org/10.7554/eLife.35471.029), using the following rationale. If the *IRA2* hotspot's effect on growth in the presence of hydrogen peroxide is partitioned into the effect due to the expression level of a given gene $i$, and the effect independent of the expression of Gene $i$ (Figure 7A), we can use the following system of linear models:

$$Gene_i = \alpha * Genotype_{IRA2} + e1$$

$$Growth_{H2O2} = \beta * Genotype_{IRA2} + \gamma * Gene_i + e2$$

Here, $Gene_i$ is the vector containing the expression values of gene $i$ for 979 segregants. $Genotype_{IRA2}$ is the vector containing the genotype of 979 segregants at the peak marker of the *IRA2* hotspot. $Growth_{H2O2}$ is the vector containing the colony growth measurements of 979 segregants in presence of hydrogen peroxide. $\alpha$ is the estimated effect of the *IRA2* hotspot on the expression of $Gene_i$. $\beta$ is the estimated direct effect of the *IRA2* hotspot on growth in presence of hydrogen peroxide that is independent of the expression level of the given gene, and $\gamma$ is the estimated effect of the expression level of $Gene_i$ on growth in hydrogen peroxide. $e1$ and $e2$ are the residual errors. From these equations, the growth as a function of the genotype at the *IRA2* hotspot marker can be expressed as follows.

$$Growth_{H2O2} = \beta * Genotype_{IRA2} + \gamma * \alpha * Genotype_{IRA2} + e2 + e1$$

$$Growth_{H2O2} = (\beta + \alpha * \gamma) * Genotype_{IRA2} + e2 + e1$$

Hence, the estimated total effect of the genotype at the *IRA2* hotspot marker on growth in the presence of hydrogen peroxide is $\beta + \alpha*\gamma$. This includes the estimated direct effect of the genotype, $\beta$, which is independent of the expression of $Gene_i$, and the effect $\alpha*\gamma$, which is mediated by the expression level of $Gene_i$. The proportion of the hotspot's effect mediated by the expression of the gene is therefore the ratio of the mediated and total effect.

$$Proportion\ mediated = \frac{Estimated\ mediated\ effect}{Estimated\ total\ effect} = \frac{\alpha * \gamma}{\beta + \alpha * \gamma}$$

We calculated the *p*-value for the presence of a mediation effect of the gene's expression on growth using 1,000 non-parametric bootstraps, the default number of simulations in the package. The FDR was estimated using the '*fdr*' method in the *p.adjust* function.

