## [Data S1. Transparent peer review records for Renganaath and Albert · Cell Genomics]

Summary

Initial submission: Received : 7/9/2024

Scientific editor: Laura Zahn

First round of review: Number of reviewers: 3
Revision invited : 10/2/2024
Revision received : 2/11/2025

Second round of review: Number of reviewers: 3
Accepted : 4/9/2025

Data freely available: Yes

Code freely available: Yes

This transparent peer review record is not systematically proofread, type-set, or edited. Special characters, formatting, and equations may fail to render properly. Standard procedural text within the editor's letters has been deleted for the sake of brevity, but all official correspondence specific to the manuscript has been preserved.

Referees' reports, first round of review

Reviewer #1: Renganaath and Albert perform valuable modeling in yeast of genetic effects on 46 growth traits and how these growth traits may be mediated by gene expression regulation, most often through trans-acting mechanisms. The rationale for comparing gene expression and growth traits in the same segregants grown and measured at different times and with different plating patterns to estimate genetic effects was well explained and justified. The evidence presented that growth traits are polygenic with most expression effects on growth acting in trans is convincing and matches human studies most expression-mediated h^2 is in trans rather than cis. My comments include suggestions for clarifying a few technical details, but the paper is very strong as currently written. Clarifications and additions that could improve the paper include:

1. Please clarify whether each FDR and Bonferroni adjustment calculation mentioned in the text used the number of comparisons/tests within a growth trait or across all 46 growth traits. Across all is more conservative than within a growth trait and should be clearly noted.
2. If you add the variance explained by trans eQTLs (median of 6 per gene) to Fig 5A, how do trans eQTL effects compare to trans eQTL hotspot effects across growth traits? Since most trans eQTLs are in hotspots, I assume it's similar? This could help quantify the effect of trans-eQTL hotspots in comparison to the strongest trans-eQTL effects.
3. I suggest including colocalization analyses for top trans-eQTL (LOD>10 or slightly more relaxed since trans effects are weaker than local) marker regions. This would help identify the most likely causal hotspots for particular growth traits. Are the "causal" hotspots the same? Do they differ across growth traits?
4. In your GO-Slim analyses, you divide genes into positive or negative correlation with growth groups before testing for GO term enrichment. In the methods, you mention "When a GO-term was significant in both the "positive" and "negative" enrichment analyses, we retained the GO-term in the analysis with the greater magnitude of log-fold enrichment." Can anything more be learned by not first dividing significant genes into positive and negative sets? For example, do particular biological processes have stronger enrichments when both directions of effect are allowed, e.g., the BP pathway includes activators and repressors? Do your genetic correlation results support what is known about how the enriched genes function in a particular pathway and its respective growth trait?
5. Could you add to the discussion how your findings fit (or don't) with the models of eQTLs and GWAS traits proposed in Mostafavi et al. 2023 (ref 24)?

Reviewer #2: This study integrated transcriptomes and several genetically complex growth traits in a large cross between two strains of *Saccharomyces cerevisiae*. The authors found that genetic correlations between expression and growth phenotypes were mostly attributable to multiple independent trans-acting eQTLs (and not local regulatory variation). The authors identified the effect of trans-acting regulatory hotspots in determining complex traits by modulating cellular states. The findings and implications are important.

In my view, the study is significant for its focus on understanding the genetic correlation between a molecular trait such as gene expression and an organism-level trait such as growth rate. However, I have some major concerns on the analyses presented in the study:

1. Data Preparation: The approach to correct gene expression for the batch and growth covariates (equation in line 531) assumes that batch and growth covariates are not correlated. Did the authors confirm this?
2. Did the authors consider shared factors underlying gene expression as in principal component analysis as potential sources of technical variation? It would be important to show much variance is explained as a function of the number of principal components from the gene expression.
3. On the simple Pearson correlation for quantifying genetic correlation between expression and growth: a) are the two phenotypes (nearly) normally distributed? b) are there not outliers in these variables?

4. Does the presence of a detectable local eQTL correlate with the expression level of the (expressed) genes?
5. Does the significant overlap of the growth QTLs with a local eQTL take into account the gene expression level of the target gene? More generally, I am concerned about the enrichment analyses throughout the paper using "random sets" without considering potential sources of confounding (e.g., LD, expression level etc.).
6. The statistical criteria for colocalization of a gQTL and an eQTL is unclear. Do the colocalization tests get adjusted for multiple hypothesis testing?
7. (lines 197-206): There appear to be traits with a **negative** correlation between the magnitudes of the significant QTL effect correlations and their matched genetic correlations. How would the authors interpret these results?
8. The estimation of the fraction of trait variance in the 46 growth phenotypes explained by DNA variation at the trans-eQTL hotspots is extremely problematic without careful clarification. For example, how would the authors interpret the result that random sets explain 28.6% of the total additive heritability? Wouldn't that indicate a massive inflation in the estimates? How could such random sets generate so much heritability? Whether due to linked non-hotspot variants or not, this would indicate seem to indicate an inflated estimate of heritability.
9. It would be important to consider the standard errors for the estimates of heritability. It's unclear how noisy the estimates are.
10. I am concerned about the various correlation analyses presented here. For example, consider figure 5B, showing a scatterplot of hotspot effect correlation coefficient against the genetic correlation coefficient. The authors present the rho and p-value. These two parameters being compared were estimated in the same dataset. Thus, the estimated strength of the agreement rho could be highly inflated.

Reviewer #3: In this study, Renganaath and Albert explore how genetic variability in gene expression impacts complex traits using the yeast model system. They find that trans-eQTL hotspots, defined as loci which impact the expression of numerous other genes, play major roles in complex traits by altering expression of numerous genes involved in key growth and stress response pathways (i.e. cellular states). This is timely as there has been a conundrum in the GWAS field as to why so few GWAS loci overlap cis-eQTL, especially when most causal variants in the human genome are thought to be non-coding variants that impact mRNA levels. A strength of the current study is that the authors perform rigorous statistical analysis to support their conclusions and carefully control for potential confounding factors. They also highlight several interesting examples (e.g. CHS6, HSP12, IRA2) that showcase potential mechanisms of how the loci impact traits. However, there are some weaknesses. It is unclear what fraction of the genetic correlations are causal for the phenotypes versus merely correlative. Furthermore, the analysis on clustering the traits by gene expression correlations into 3 groups seems over-simplified and needs some followup. Overall, the study should be of interest to Cell Genomics readers and suitable for publication if the authors can adequately address these points.

Major comments:

It is unclear what fraction of the genetic correlations are relevant to the phenotypes. For example, could it be that most trans-eQTL hotspots mediate most of their phenotypic effect through a couple genes for each condition? One way to address this concern would be to look at which genes are significantly down-regulated by a hotspot and check whether deletion or CRISPRi repression of those genes in previous screens had a phenotypic effect. These data are probably unavailable for all conditions in this current study but even a couple conditions would help address this point.

In Figure 4A, some hotspots show a QTL effect in almost all conditions, for example signals on chrXIV and chrXV. Does the genetic correlation analysis suggest that the same target genes are important in all these conditions?

The authors report that trans-eQTLs played a much greater role in the genetic correlations than the local eQTLs. Is this simply a consequence of the trans-eQTLs having a greater impact on the expression of the gene collectively? One would expect that regardless of whether the eQTL were cis or trans, the consequence

of regressing out its impact on growth should be related to the magnitude of the effect on expression.

How do the 210 gQTLs which do not colocalize with a trans-eQTL hotspot differ from the 398 gQTLs? Do they differ in effect size, direction, or the impacted conditions?

The authors cluster the traits into two main groups (named green and blue) based on the genetic correlations (page 19). In the green group, they find GO term enrichments for ribosome biogenesis for genes with positive correlations and cell cycle control and various stress responses for negative correlations. The blue group shows a reverse trend. The authors then make a comparison with genes associated with faster and slower chemostat-based growth in a previous study. This is an interesting observation but the analysis seems a bit shallow and perhaps oversimplified. Overall, do these green and blue growth conditions target distinct pathways such that this grouping makes sense? How do these conditions cluster based on the segregant growth data alone? It appears that most conditions involve some sort of stress in the form of metal ion or chemical treatment, yet stress conditions appear in both the green and blue groups. For example, cisplatin is in the blue group while 4-NQO is in the green group. As both are DNA damaging agents, it is surprising that these are not clustered together.

Minor comments:

The manuscript could benefit from some streamlining. There are many redundant points made in the intro, results and discussion. For example, it is unnecessary for the authors to repeatedly mention that the eQTL and gQTL studies were done at different research institutions.

It is unclear what point the authors are trying to make in the results section titled "Most growth QTLs are colocalized with multiple local eQTLs". First they mention that "all but one of the 591 gQTLs in our dataset overlapped with at least one local eQTL, slightly but significantly more than expected by chance". To place this into context, it would help if the authors stated this number (i.e. the number of gQTLs that would be expected to overlap local eQTL by chance).

It could be helpful to clear up the numbers for the eQTLs and the transcripts they impact. Fig 1 states that 5648/5720 genes have at least one eQTL. On line 108, the authors state "5,643 genes that have at least one eQTL in liquid YNB medium". Do the authors mean that 5,643 transcripts (or 5,648 - need to clear up what the exact number is), which is essentially all transcripts, have their levels impacted by an eQTL somewhere in the genome? Or do they mean that 5,643 genes have eQTLs mapped down at the gene-level, meaning that nearly every gene impacts the levels of a transcript somewhere in the genome?

Fig 7B. It is difficult to distinguish between the red and gray circles due to the dense overlap. Perhaps these data can be shown in a different way to better separate Msn2 targets from non-targets, such as jittering the points or faceting the plot.

The concept that "inherited alleles shape an individual's gene expression profile in a baseline condition in ways that can influence growth when cells encounter additional environmental factors" has been previously explored (e.g. see Gagneur et al. 2013, PMC3778020) This previous study showed that expression variation that is relevant to a phenotype tends to be persistent across different conditions. While the current study only looked at eQTLs in a single condition (YNB), this is worth mentioning in the discussion. This is also relevant to this sentence on line 127, page 5. "These effect sizes suggest that genetic effects on gene expression in a standard environment shape growth in different environments partially but not completely, resembling recent results in humans."

Authors' response to the first round of review

In this document, reviewer comments are given in black Times New Roman Font, and our responses are given in blue Arial font.

Reviewer #1

Renganaath and Albert perform valuable modeling in yeast of genetic effects on 46 growth traits and how these growth traits may be mediated by gene expression regulation, most often through trans-acting mechanisms. The rationale for comparing gene expression and growth traits in the same segregants grown and measured at different times and with different plating patterns to estimate genetic effects was well explained and justified. The evidence presented that growth traits are polygenic with most expression effects on growth acting in trans is convincing and matches human studies most expression-mediated h^2 is in trans rather than cis. My comments

include suggestions for clarifying a few technical details, but the paper is very strong as currently written.

We thank the reviewer for this positive assessment.

Clarifications and additions that could improve the paper include:

1. Please clarify whether each FDR and Bonferroni adjustment calculation mentioned in the text used the number of comparisons/tests within a growth trait or across all 46 growth traits. Across all is more conservative than within a growth trait and should be clearly noted.

FDR corrections were all performed within-trait. We have added this information to the paper at the relevant Results sections as well as the Methods.

2. If you add the variance explained by trans eQTLs (median of 6 per gene) to Fig 5A, how do trans eQTL effects compare to trans eQTL hotspot effects across growth traits? Since most trans eQTLs are in hotspots, I assume it's similar? This could help quantify the effect of trans-eQTL hotspots in comparison to the strongest trans-eQTL effects.

The result of the analysis suggested by the reviewer is shown below, with the variance explained by all trans-eQTL peak markers added as teal points (Reviewer Figure 1). As can be seen, these estimates closely track those for the entire genome, estimated using all markers. The reason for this result is simply that the set of all trans-eQTLs (5,149 markers across the 5,605 genes with at least one trans-eQTL) is so large that it is in LD with nearly the entire genome (represented by 11,530 markers in our analyses), such that the heritability from all trans-eQTLs approaches the genome-wide heritability.

Response to Reviewers

The reason the reviewer's intuition that the heritability from all trans-eQTLs should be close to that from the hotspots (as opposed to the actual observation that it is closer to the entire genome) is not met is due to two reasons. First, while most trans-eQTLs are at hotspots, not all of them are. Each of the additional, non-hotspot, trans-eQTLs adds a marker to the analysis below. Due to the way the 102 hotspot locations were estimated in Albert, Bloom, et al. 2018, these added markers are not in close linkage with the 102 hotspots. Second, many of the individual trans-eQTLs that do correspond to a hotspot have peak markers that are located some distance away from the given hotspot marker rather than being exactly at the hotspot marker. Note that the hotspots, just like regular QTLs, have positional uncertainty due to linkage; the 102 markers reported in Albert & Bloom et al. correspond to the most likely estimates of hotspot positions across all the genes with trans-eQTLs at the given position. Thus, these trans-eQTLs widen the region of the genome that is in LD with the hotspots compared to using only the 102 hotspot markers. Together, the addition of these two sets of markers (unlinked to hotspots and scattered in the vicinity of the hotspots) adds much of the genome to the heritability estimates, which naturally approaches the whole-genome estimate. We have chosen not to include this analysis in the revised paper as we don't feel that the result adds conceptually to the paper, but are grateful to the reviewer for the prompt to explore this question.

Reviewer Figure 1: Proportion of phenotypic variance explained by all 11,530 genetic markers (narrow-sense heritability; red dots), the 102 trans-eQTL hotspots (blue dots), all trans-eQTLs (teal dots), and all gQTLs (yellow dots), in 46 growth conditions. Error bars about each estimate (depicted in the same color as the estimates) represent the bootstrapped standard errors computed via 5,000 bootstraps. Box plots show the distribution of variance explained by 1,000 sets of 102 random genetic markers. These distributions were used to calculate p-values (shown at top) for the proportion of variance explained by the hotspots. *: $p < 0.05$.

3. I suggest including colocalization analyses for top trans-eQTL (LOD>10 or slightly more relaxed since trans effects are weaker than local) marker regions. This would help identify the most likely causal hotspots for particular growth traits. Are the "causal" hotspots the same? Do they differ across growth traits?

The analysis suggested by the reviewer would involve 77,562 colocalization tests at the suggested LOD threshold, a considerable increase compared to the 2,078 tests currently in the paper. We are not certain that these computationally intensive analyses would add substantially to the paper.

Instead, we performed a different analysis to address the reviewer's question about which hotspots are likely to be causal for each trait. To do so, we revisited the analyses that were shown in Supplementary Figure 9 as well as Figure 4D in our initial submission. In these analyses, we asked how much variance was explained by hotspot sets of increasing size, with hotspots ranked by the number of genes each hotspot affects. Across the 46 traits, Supplementary Figure 9 had shown that there is considerable heterogeneity in how much of an increase a given hotspot causes for a given trait.

We have now extended this analysis by quantifying what fraction of the total variance explained by all hotspot markers in a given trait is contributed by each hotspot as it is added to the

analysis. A hotspot that is causal for a trait (or closely linked to a causal gene) should contribute a comparatively large amount of variation. The results again show heterogeneity in which hotspot affects which traits. Some hotspots contribute substantially to several traits (e.g. hotspots at chrII:541139, which affects growth on several carbon sources, and at chrI:159467, which affects growth on several stressors), while other hotspots have more trait-specific contributions (such as a hotspot at chrIV:156943 that has particularly strong contributions to growth on cycloheximide). No causal genes have been identified for these three hotspots, and their connections to diverse growth traits make them attractive targets for future dissection. In the revised manuscript, we have added the results above along with the figure below as a new Supplementary Figure 10 with associated Table S6 .

Reviewer Figure 2 (included now as Supplementary Figure 10): Proportion of phenotypic variance explained by each of the 102 trans eQTL hotspots detected in Albert et al. (2018) for the 46 growth traits.

4. In your GO-Slim analyses, you divide genes into positive or negative correlation with growth groups before testing for GO term enrichment. In the methods, you mention "When a GO-term was significant in both the "positive" and "negative" enrichment analyses, we retained the GO-term in the analysis with the greater magnitude of log-fold enrichment." Can anything more be learned by not first dividing significant genes into positive and negative sets? For example, do

particular biological processes have stronger enrichments when both directions of effect are allowed, e.g., the BP pathway includes activators and repressors? Do your genetic correlation results support what is known about how the enriched genes function in a particular pathway and its respective growth trait?

We thank the reviewer for their suggestion. While addressing this comment, we noticed an error in our code, which had led to the incorrect inclusion of depleted (in addition to enriched) GO categories in this section. Correcting this led to minor changes in which traits were assigned to the three main trait groups but did not alter our overall conclusions. In fact, the contrast between the “green” and “blue” group is now sharper. We have updated the Figure 6 and main text accordingly.

In direct response to the reviewer’s comment, we performed GO enrichment analyses without binning genes into positive and negative genetic correlation sets. The resulting heatmap is shown in the figure below (Reviewer Figure 3). Comparing these GO term enrichment results to the (updated) direction-aware results in the paper showed that the GO-BP term enrichments were significantly weaker (using Wilcoxon tests comparing log-fold enrichment of each GO category and a p-value threshold of 0.05) for 41 out of 46 conditions when ignoring the direction of the genetic correlation (Reviewer Figure 4). Approximately 83% of significant enrichments across all 46 conditions became non-significant. Furthermore, 87% of significant enrichments that were detected without considering direction had also been detected when performing the initial enrichment analysis separated by direction.

Given that the GO enrichments from the analysis that combined both directions of genetic correlations were largely captured by our initial analysis and given that our initial analyses produced stronger and more numerous GO term enrichments, we did not include the direction-free analyses in the manuscript. We do thank the reviewer for their prompt to explore this question.

Reviewer Figure 3: Biological processes enriched in genetic correlations with GO term enrichment done ignoring the direction of genetic correlations of the genes. As in Figure 6, the heatmap shows log₂-fold enrichments for 100 GO-BP Slim terms in the set of genes with significant (5% FDR) genetic correlations. Enrichments are displayed for terms significant at uncorrected $p < 0.001$. The bar plot above the heatmap shows the number of genes with significant genetic correlation (5% FDR) for each condition. We here cluster the conditions again into three groups to maintain consistency with the analysis in Figure 6, but indicate the groups here by different colors to underscore the different groupings of traits in both the analyses.

Reviewer Figure 4: Paired box plots depicting the changes in magnitude of log₂ fold enrichments of the 100 GO-BP terms for the 46 growth traits, when considering the directions of genetic correlations before GO enrichment vs not considering the directions. The p values from paired Wilcoxon tests are indicated for each trait.

5. Could you add to the discussion how your findings fit (or don't) with the models of eQTLs and GWAS traits proposed in Mostafavi et al. 2023 (ref 24)?

We thank the reviewer for this suggestion. A central insight from the excellent work by Mostafavi et al. is that human eQTL mapping and GWAS for complex traits tend to detect different genes

7

(and different variants at the same genes) due to differences in statistical power for eQTL

mapping and GWAS, especially in the presence of negative selection. The Mostafavi et al. paper is clearly conceptually relevant for our work here, but many differences in the data underlying their and our papers make direct comparisons challenging. For example, our data lack the spatial resolution to make statements about the proximity of causal variants to their genes for eQTLs vs gQTLs. The difference in power to detect eQTLs versus gQTLs is much smaller in our data than the corresponding difference noted by Mostafavi et al. in humans: while our eQTLs may be somewhat less powered than our gQTLs due to multiple testing (there are more expressed genes than there are measured traits), the sample size is the same in both yeast studies, such that any power difference is small. An assumption in Mostafavi et al. that any trans-eQTLs and variant effects on complex traits ultimately arise from cis-eQTLs is also clearly not met in this yeast cross, in which coding variants are known to underlie many (and perhaps most, at least based on current sets of known genes) gQTLs and trans-eQTL hotspots. Perhaps most crucially, in contrast to Mostafavi et al., we lack a sizeable set of known or likely causal genes at gQTLs, which are usually too wide to be able to use the genes closest to the peak as performed by Mostafavi et al. The smaller number of such genes than are available at human GWAS signals precludes meaningful comparisons of features of the genes with local eQTLs (which is half the genes in our data) and genes that are causal for complex growth phenotypes.

With these caveats in mind, Mostafavi et al.'s central insight that cis-eQTLs are depleted at genes that influence complex traits is mirrored by our conclusion that "many local eQTLs appear to have little influence on growth traits because they do not alter the relevant central processes". We have added another reference to Mostafavi et al. to this point in the Discussion, and have added that this observation may be because more directly functional cis-eQTLs may tend to be removed by negative selection. We prefer to leave deeper and more direct comparisons between human and yeast data to future work.

Reviewer #2

This study integrated transcriptomes and several genetically complex growth traits in a large cross between two strains of *Saccharomyces cerevisiae*. The authors found that genetic correlations between expression and growth phenotypes were mostly attributable to multiple independent trans-acting eQTLs (and not local regulatory variation). The authors identified the effect of trans-acting regulatory hotspots in determining complex traits by modulating cellular states. The findings and implications are important. In my view, the study is significant for its focus on understanding the genetic correlation between a molecular trait such as gene expression and an organism-level trait such as growth rate. However, I have some major concerns on the analyses presented in the study.

We thank the reviewer for this overall positive assessment, and address all concerns below.

8

1. Data Preparation: The approach to correct gene expression for the batch and growth covariates (equation in line 531) assumes that batch and growth covariates are not correlated. Did the authors confirm this?

We thank the reviewer for this question, but wish to point out that we do not make an assumption that batch and growth covariates are uncorrelated. The linear model referred to should remove all OD and batch effects even if there is correlation between these two gene expression covariates. Reviewer Figure 5 shows the OD values for each of the 13 batches.

Reviewer Figure 5: Boxplot of the Growth (OD) covariates for different batches of segregants in Albert et al. (2018)

Note that there are some differences in OD between the batches, likely due to how the samples were grown in Albert & Bloom et al., 2018. There, the segregants had been grouped into batches (i.e., 96-well plates) based on their colony sizes on solid YNB medium in Bloom et al. (2013). The different 96-well plates were then grown for slightly different times, aiming for similar average OD values across plates. Evidently, that procedure in Albert & Bloom et al. (2018) was somewhat imperfect, as batches did differ slightly in their average ODs. (As a side note, the batch IDs, while being shown as numbers, do not correspond to numeric values; they are treated as a categorical variable by the linear model. We have made this more explicit in the revised methods, which now explicitly state that batch was “modeled as a categorical factor”).

9

The observed difference in OD values among batches is one reason why “batch” was corrected out here and in Albert & Bloom et al., 2018. The OD covariate then corrects for any residual effects of different OD of individuals within each batch.

Also note that the growth covariate corrected out here (OD600 in liquid growth at the time the cultures were harvested for RNA isolation) is different from the colony size traits used as complex growth traits throughout this paper.

2. Did the authors consider shared factors underlying gene expression as in principal component analysis as potential sources of technical variation? It would be important to show much variance is explained as a function of the number of principal components from the gene expression.

We thank the reviewer for this prompt to examine principal components in the gene expression data. The figure below shows the fraction of variance explained by these principal components (PCs):

Reviewer Figure 6 (now included as Supplementary figure 15): Cumulative distribution of the proportion of variance among gene expression (for 5720 genes assayed in Albert et al.) explained by principal components. The inset shows a scree plot for the same principal component analysis. For principal components with Eigenvalue ≥ 1 , points are indicated in dark pink.

As can be seen, hundreds of PCs are needed before Eigenvalues drop to less than one, and 63 PCs are needed to account for 80% of the variance. These large numbers of PCs demonstrate considerable complexity in the transcriptomes analyzed here. We further examined the first 10 PCs, which together accounted for 65.70% percent of gene expression variation:

PC	Proportion of Variance
1	0.30211
2	0.14144
3	0.0578
4	0.04123
5	0.03157
6	0.02171
7	0.01954
8	0.01596
9	0.01435
10	0.01115

We first asked if biological processes are enriched among the genes with the strongest loadings, separately for positive and negative loadings (Figure below). Numerous enrichments were found. Because technical variation would not be expected to specifically target certain biological processes, these enrichments suggest that these PCs do not reflect technical variation but instead reflect biological variation among individuals, due to both genetic influences with widespread effects such as trans-eQTL hotspots and random biological fluctuations. We have added this information to the Methods section (Page 32, lines 549-553) and added Reviewer Figures 6 and 7 to the paper as Supplementary Figures 15 & 16.

Reviewer Figure 7 (now included as Supplementary Figure 16): Biological processes enriched in expression PCs. The top heatmap shows significant \log_2 -fold enrichments (uncorrected $p < 0.001$) for 100 GO-BP Slim terms in the top 10% set of genes that load most strongly in both directions (negative indicated by blue and positive indicated in red). The heatmaps at the bottom show average PC loadings for genes in the indicated gene groups from Brauer et al. and Gasch et al.

Further examination of the biological processes enriched in the PCs showed that the first two principal components (which account for 30% and 14% of variation, see table above) correspond closely to the major cellular states we describe in the paper. This is especially true for PC2, which, based on its enriched GO terms, reflects a balance between translation and carbohydrate metabolism and the stress response. PC2 scores are also correlated with gene expression in the ESR and in slow versus fast growth responses (see bottom of the Figure above). These patterns in PC2 closely match those seen in the "green" trait group we described in Figure 6. Indeed, there is very strong correlation between the GO enrichments for PC2 and those for traits in the "green" group such as lactose, YNB:pH8, or tunicamycin):

Reviewer Figure 8: Heatmap showing the correlation between the GO enrichment patterns (expressed as fold-change enrichments for each GO term) for each trait with each of the first 10 PCs.

Reviewer Figure 8: Heatmap showing the correlation between the GO enrichment patterns (expressed as fold-change enrichments for each GO term) for each trait with each of the first 10 PCs.

The first principal component PC1 is enriched for terms related to DNA metabolic processes, DNA repair, and the cell cycle. These terms are also enriched in a subgroup of the “blue” trait group in Figure 6 (including, for example, formamide, 4-hydroxybenzaldehyde, and menadione). These traits also show marked correlations between their GO enrichments and the GO enrichments for PC1 (Reviewer Figure 8, top right corner).

Together, these observations suggest that 1) the transcriptomes analyzed here are heterogeneous, 2) their major axes of variation reflect biological variation among individuals in broad cellular processes, 3) the biological variation picked up by the PCs corresponds, at least in part, to the same major cellular states we describe in this paper. In the interest of brevity, we have chosen not to add these direct comparisons of PCs to cell states and trait groups to the paper.

3. On the simple Pearson correlation for quantifying genetic correlation between expression and growth: a) are the two phenotypes (nearly) normally distributed? b) are there not outliers in these variables?

We thank the reviewer for this comment. The following figure displays QQ-plots for all 46 growth traits against a normal distribution, showing that all traits are well approximated by normal

distributions:

Reviewer Figure 9: QQ- plots comparing the distribution of growth values of 979 segregants from Bloom et al. against a normal distribution.

Reviewer Figure 10 shows the same information for expression values of 100 randomly selected genes. Most genes follow a normal distribution fairly well, with no strong outliers:

Reviewer Figure 10: QQ-plots comparing the distributions of expression values of 979 segregants for 100 randomly selected set of genes against a normal distribution.

These observations suggest that the Pearson correlations we used in the paper are reasonable estimates of the correlation between gene expression and growth. We also note that the QTLs mapped in Bloom et al. 2013 and Albert & Bloom et al., 2018 used linear models that assume normality, such that our work here is in line with these prior publications on the same datasets.

16

4. Does the presence of a detectable local eQTL correlate with the expression level of the (expressed) genes?

To address this question, we divided genes into those with and those without a local eQTL and compared their expression levels. As the figure below shows, there is no difference in expression level between these two groups:

Reviewer Figure 11: Comparison of median mRNA expression of genes with and without local eQTLs. mRNA expression values considered are not corrected for batch and growth effects, and their median for the respective genes are computed across the complete set of segregants assayed in Albert et al. (2018)

The corresponding medians are extremely similar as well:

Has at least one local detectable eQTL in Albert et al. (2018)	Median corrected median mRNA expression (batch and OD corrected) across 1012 segregants	Median uncorrected median mRNA expression (batch and OD corrected) across 1012 segregants
TRUE	0.514	5.7
FALSE	0.517	5.7

5. Does the significant overlap of the growth QTLs with a local eQTL take into account the gene expression level of the target gene? More generally, I am concerned about the enrichment analyses throughout the paper using "random sets" without considering potential sources of confounding (e.g., LD, expression level etc.).

The specific analyses the reviewer asked about (overlap of the growth QTLs with a local eQTL) do not take into account the expression level of the target gene. Based on the analyses in response to comment 4 above, which showed no difference between genes with and without a local eQTL, there is no need for this analysis, nor is there a way to formally incorporate expression level into the colocalization test.

To address the reviewer's more general concern about random sets being poor matches for the various actual sets, we focus on three major analyses in the paper that use random sets and

that exemplify that our use of random sets is appropriate: (1) Overlap between gQTLs and eQTL hotspots, (2) Correlation between genetic and hotspot effect correlations, (3) The heritability estimates in Figure 5A. For the first analysis, we do not see how expression level could be considered because each hotspot affects the expression of multiple genes, each with a different expression level. Likewise, there is no way to consider expression level in the third analysis. For the second analysis, we stratified the genetic correlation results into five bins with equal numbers of genes with increasing expression level, based on their average expression across segregants:

Reviewer Figure 12 (now included as Supplementary Figure 12): Spearman correlation coefficients between hotspot effects and genetic correlation coefficients across all conditions and genes, stratified by expression quintiles. Violin plots illustrate the distribution of correlation coefficients for 1,000 randomly generated sets of 102 markers. The actual correlation coefficients for the 102 hotspot markers are represented as pink dots. (Note: The result for "all genes" corresponds to Figure 5B.) The X-axis also indicates whether the mean correlation coefficients for the random sets within each quintile bin are significantly different from the "all genes" set, as determined by Wilcoxon tests ($p < 0.05$): ns = not significant, * = significant.

Spearman correlation coefficient between
the hotspot effect and genetic correlation coefficients

As is apparent from this figure, even though there are tiny (note the scale on the lower part of the y-axis) albeit significant differences in the means of the distributions for the random sets for four of the quintile bins compared to all genes, the correlation between hotspot effects and genetic correlations for the actual 102 hotspot markers greatly exceeds those for the random marker sets in every quintile. We conclude that expression level is not an important confounder of these analyses. We have added this information to the paper as Supplementary Figure 12 and at page 18, lines 287-288.

To address the reviewer's concern about the random sets not matching LD patterns of the actual positions, which could affect analyses (1) and (3) above, we first note that in the BY/RM cross, LD is highly uniform along the genome. There are no recombination hotspots as are seen in human populations and that the reviewer may be familiar with. Indeed, this uniform recombination rate is one of the reasons this cross was initially chosen for complex trait dissection more than two decades ago. In addition, the 102 hotspots are in LD with much of the genome (see below in our response to comment 8 from this reviewer), such that any subtle recombination rate variation is likely averaged out for such a large set of markers. For these reasons, we do not expect variation in LD along the genome to affect our results.

To confirm this expectation, we computed the number of markers that are in LD (i.e., correlation of marker genotypes), with the 102 hotspots at different LD thresholds, both for the actual set of 102 hotspots and for the random sets of 102 markers. Our goal was to ask how much of the genome is in LD with the given marker sets.

Reviewer Figure 13 (now included as Supplementary Figure 8): Boxplots comparing the number of markers in LD with the 102 trans eQTL hotspot markers and 102 random markers, at the different LD ranges indicated on the x-axis. The values for the light gray boxes are the median of the number of markers for each of the 1000 random sets of 102 markers in the genome at the indicated LD ranges. LD between two markers is computed as the Pearson correlation coefficient between the genotypes at both markers.

As expected, an increasing number of markers are in LD with the 102 hotspots as the LD threshold decreases, with up to 100 markers for the median hotspot marker at LD > 0.6. Crucially, the number of markers in LD with the random sets closely match those of the actual 102 hotspots. To the extent that there is any difference, the random sets tag more markers than the actual hotspots. As such, the hotspots would be expected to capture less heritability than random sets. This makes the analyses in Figure 5A conservative, and strengthens our observation that the hotspots account for more variation in growth phenotypes than expected by chance. Overall, we conclude that differences in LD between the actual markers and the random sets do not meaningfully bias our results.

We have added the figure on LD results above to the paper as Supplementary Figure 8 and state the conclusion of the LD results in the main text (Page 13, lines 239-240).

21

6. The statistical criteria for colocalization of a gQTL and an eQTL is unclear. Do the colocalization tests get adjusted for multiple hypothesis testing?

The colocalization analyses were deliberately not corrected for multiple testing. This was done because in these analyses, a significant p-value leads to rejection of the pleiotropic null model when a model with two independent QTLs fits the data better than a model with a single

pleiotropic QTL. Our goal in this section is to identify QTLs that are pleiotropic (i.e., those for which the test is not significant), because at these loci the eQTL and the growth QTL could arise from the same causal variants. As such, using a liberal criterion for rejecting pleiotropy, without correcting for multiple testing, is conservative in this particular instance. The Methods contain information on this point, and we have updated the Results section (page 7, lines 147-149) to further clarify this point.

7. (lines 197-206): There appear to be traits with a **negative** correlation between the magnitudes of the significant QTL effect correlations and their matched genetic correlations. How would the authors interpret these results?

We first note that there are only two traits with significant negative correlations (uncorrected for multiple tests) in this analysis: formamide and lithium chloride, with p-values of 0.02 and 0.002, respectively. The remaining traits do not show significant correlations between QTL effects and genetic correlations. Crucially, the correlations for these two traits are based on only a handful of genes (6 and 3, respectively). Our interpretation of their nominally significant negative correlations is simply that they are likely to be false positive traits in an analysis that is overall not well powered, as we pointed out in the manuscript.

8. The estimation of the fraction of trait variance in the 46 growth phenotypes explained by DNA variation at the trans-eQTL hotspots is extremely problematic without careful clarification. For example, how would the authors interpret the result that random sets explain 28.6% of the total additive heritability? Wouldn't that indicate a massive inflation in the estimates? How could such random sets generate so much heritability? Whether due to linked non-hotspot variants or not, this would indicate seem to indicate an inflated estimate of heritability.

The reason for this apparent "inflation" is the high degree of linkage in this one-generation cross. Starting with two haploid genomes, the segregants analyzed here are the product of one round of meiosis. Crossing over in yeast occurs at a fairly high and mostly uniform rate, such that each recombinant progeny usually has a handful of breakpoints between the parental genomes on each chromosome. However, linkage in the resulting panel is still quite strong, such that each genetic marker is linked to a sizable portion of the neighboring genome. To illustrate this point quantitatively, the table below shows the average number of markers that 102 random variants are linked to at various thresholds of LD, along with the fraction of all markers in the genome that this corresponds to:

LD range	Median number of markers in given LD range with 102 random markers	Median fraction of genome in given LD range with 102 random markers
> 0.7	8,806	0.76
> 0.8	5,997	0.52
> 0.9	3,151.5	0.27

As can be seen, even at a strict LD threshold of 0.9, 102 randomly picked markers tag 27% of all markers. Therefore, it is not surprising that these markers pick up a non-trivial fraction of the heritability of a polygenic trait by chance. Indeed, the reason why we conducted analyses of random marker sets in the paper is to control for this high degree of background linkage. We have further clarified this point throughout the paper, for example at lines 234 and 721-723.

9. It would be important to consider the standard errors for the estimates of heritability. It's unclear how noisy the estimates are.

We computed standard errors of the heritability estimates by performing a bootstrap resampling procedure with 5,000 iterations. The median standard errors across 46 conditions for each of the estimates are summarized below. As the table and Reviewer Figure 14 (below) shows, the standard errors are small and do not affect our conclusions.

Genomic markers considered	Median heritability estimate	Range of heritability estimates	Median standard error	Range of standard errors

All genetic markers	0.58	0.28-0.82	6e-09	1e-09 - 5e-03
All trans eQTL markers	0.54	0.26-0.93	6e-09	0 - 1e-02
102 trans-eQTL hotspots	0.43	0.04-0.62	0.02	0.019 - 0.04
All gQTLs of a condition	0.45	0.06-0.65	0.02	0.015 - 0.05

Reviewer Figure 14 (included as revised Figure 5A): Proportion of phenotypic variance explained by all 11,530 genetic markers (narrow-sense heritability; red dots), the 102 trans-eQTL hotspots (blue dots), and all gQTLs (yellow dots), in 46 growth conditions. Error bars about each estimate (depicted in the same color as the estimates) represent the bootstrapped standard errors computed via 5,000 bootstraps. Box plots show the distribution of variance explained by 1,000 sets of 102 random genetic markers. These distributions were used to calculate p-values (shown at top) for the proportion of variance explained by the hotspots. *: $p < 0.05$.

We have revised Figure 5A to show the standard errors corresponding to each heritability estimate.

24

10. I am concerned about the various correlation analyses presented here. For example, consider figure 5B, showing a scatterplot of hotspot effect correlation coefficient against the genetic correlation coefficient. The authors present the rho and p-value. These two parameters being compared were estimated in the same dataset. Thus, the estimated strength of the agreement rho could be highly inflated.

To test if our results are influenced by this phenomenon, we performed 100 random splits of the data. In each split, half the segregants were used to estimate the genetic correlations and the

other half was used to estimate the hotspot effect correlations or the QTL effect correlations.

Comparison	Spearman correlation coefficient from all 1000 segregants for both correlations	Median Spearman correlation coefficient across 100 random samplings of the data using separate 50% splits for estimating the two correlation coefficients	Percentage of splits with significant correlation ($p < 0.05$) between genetic and hotspot effect correlations
Hotspot effect correlation vs Genetic correlation	0.86	0.69	100%
QTL effects correlation vs Genetic correlation	0.69	0.59	100%

While the correlations based on the random splits have smaller coefficients, they remained positive and were statistically significant in every split. Thus, our overall conclusions from these analyses remain unchanged. We have added the results above to the paper (page 10, lines 208-210 & page 18, lines 288-290).

Reviewer #3

In this study, Renganaath and Albert explore how genetic variability in gene expression impacts complex traits using the yeast model system. They find that trans-eQTL hotspots, defined as loci which impact the expression of numerous other genes, play major roles in complex traits by altering expression of numerous genes involved in key growth and stress response pathways (i.e. cellular states). This is timely as there has been a conundrum in the GWAS field as to why so few GWAS loci overlap cis-eQTL, especially when most causal variants in the human genome are thought to be non-coding variants that impact mRNA levels. A strength of the current study is that the authors perform rigorous statistical analysis to support their conclusions and carefully control for potential confounding factors. They also highlight several interesting examples (e.g. 25

CHS6, HSP12, IRA2) that showcase potential mechanisms of how the loci impact traits.

However, there are some weaknesses. It is unclear what fraction of the genetic correlations are causal for the phenotypes versus merely correlative. Furthermore, the analysis on clustering the traits by gene expression correlations into 3 groups seems over-simplified and needs some followup. Overall, the study should be of interest to Cell Genomics readers and suitable for publication if the authors can adequately address these points.

We thank the reviewer for this positive assessment, and have addressed their comments below.

Note that we have numbered the reviewer's comments.

Major comments:

1. It is unclear what fraction of the genetic correlations are relevant to the phenotypes. For example, could it be that most trans-eQTL hotspots mediate most of their phenotypic effect through a couple genes for each condition?

The reviewer is exactly right. One of our overarching conclusions is that not every single gene with a genetic correlation is individually causal. Instead, there are underlying cell states (say, globally higher translation) that are reflected in expression change at hundreds of genes (say, higher expression of the many ribosomal protein genes and other translation-related genes). In transcriptomic data, the cellular state is "visible" as the collective expression changes at many genes. It is the state as a whole that ultimately influences growth. Most individual genes only serve as markers of the state. We have added a sentence to the Discussion (page 28, lines 465-466) to make this clearer.

At hotspots that are caused by, say, a single DNA variant in a single gene, that gene is the only truly causal gene from a genetic perspective. Variation in this gene kicks off a cascade of causality that affects broad cellular states. The reviewer's question is ultimately about the

structure of this cascade – how many genes are essential for transmitting the genetic signal instead of simply being peripheral leaves that implement a state change but do not cause it. In our opinion, it is indeed likely that most individual hotspots have a small number of genes that transmit the underlying genetic effect to broader cell states. In the paper, we have attempted to illustrate this logic with our analyses of the IRA2 hotspot. Given the fairly manual nature of these analyses (e.g. knowledge of RAS signalling through Msn2), it would be challenging (and beyond the scope of this paper) to conduct similar analyses across all hotspots. Disentangling the causal cascades that emanate from hotspots is an area for future investigation.

2. One way to address this concern would be to look at which genes are significantly down-regulated by a hotspot and check whether deletion or CRISPRi repression of those genes in previous screens had a phenotypic effect. These data are probably unavailable for all conditions in this current study but even a couple conditions would help address this point.

26

As discussed in the comment above, we are not sure that there really is a “concern” here – we do not claim that every gene with a genetic correlation is causal, but that multiple correlated genes are indicative of cellular states.

To nonetheless explore this point, we compared our genetic correlations to the effects of gene deletions, using data for 27 conditions assayed here and present in Turco et al. (2023), who gathered and systematically analyzed data from yeast gene deletion screens. We polarized the data from Turco et al. such that decreased growth caused by deleting a gene (i.e., reducing its expression to zero) would correspond to a negative genetic correlation. For genes with a significant genetic correlation at an FDR of 5%, there is a positive, significant, but small correlation between deletion effects and our genetic correlations ($\rho = 0.015$, $p = 0.004$). While this result suggests that on average our results show some correspondence to those obtained with the deletion collection, their interpretation is not straightforward. The exact details (e.g. solid vs liquid media, compound concentrations,...) of the traits used to assay the yeast deletions likely differ from those used in Bloom et al. (2013). It is not clear that complete elimination of a gene product caused by a gene deletion is necessarily expected to have consistent effects with the much more subtle expression changes caused by genetic variation, due to possible nonlinear effects linking gene abundance to a trait. We also note that the yeast deletion collection harbors secondary mutations that can alter the effects of the gene deletions. In addition to these technical considerations, it is also possible that stronger agreement between our results and those from the deletion collection is prevented because our genetic variation study and experimental deletion screens pick up different kinds of genetic signals. The genes with the strongest, most specific and direct effects on a trait identified in deletion screens may not carry functional variation in our cross, either by chance or because such variants could be deleterious in natural isolates. Conversely, the genes that shape variation in these traits in natural populations such as our cross appear to (as we show here) exercise their effects indirectly by altering broad, non-specific, cellular states. These two kinds of genes may simply not be the same.

Overall, we view the marginal agreement between the deletion collection results and those in the present manuscript as an interesting topic for future work, and we thank the reviewer for the prompt to explore this question. Given the speculative nature of our interpretations above, we have chosen not to include the results above in the revised manuscript.

3. In Figure 4A, some hotspots show a QTL effect in almost all conditions, for example signals on chrXIV and chrXV. Does the genetic correlation analysis suggest that the same target genes are important in all these conditions?

We thank the reviewer for this question. While the genetic correlation analyses cannot be computed at individual hotspots (genetic correlations at a single locus are simply the effects of that locus on the given gene or trait; for the hotspots, these are reported in Albert & Bloom et al., 2018), it is possible to count for how many traits a given gene shows a genetic correlation. The

27

resulting distribution is shown below. There are indeed a number of genes that show correlations with many traits. The top group is quite heterogeneous: the top gene (APE3) encodes an vacuolar aminopeptidase (39 traits), and the three genes genetically correlated with 38 traits differ in their functions (HSP12 and GRE1 have functions in the stress-response, broadly defined, and ADH2 encodes an alcohol dehydrogenase). As explained above, given our overall interpretation of the results in terms of cellular states rather than individual genes, we prefer not to draw too much attention to individual genes in the text. Table S3 in the paper contains the underlying data for these analyses, such that interested readers can conduct their own analyses of these data.

Reviewer Figure 15: Histogram showing the distribution of genes based on the number of growth conditions in which they exhibit significant genetic correlation (at FDR < 5%)

4. The authors report that trans-eQTLs played a much greater role in the genetic correlations than the local eQTLs. Is this simply a consequence of the trans-eQTLs having a greater impact on the expression of the gene collectively? One would expect that regardless of whether the eQTL were

28 cis or trans, the consequence of regressing out its impact on growth should be related to the magnitude of the effect on expression.

The reviewer is correct. The greater contribution of the trans-eQTLs to the genetic correlations is most likely a function of there being multiple trans-eQTLs per gene. This is the same reason for why trans-eQTLs explain more of the heritability in gene expression than do cis-eQTLs: when summed up, the effects of multiple trans-eQTLs, even though weak individually, are stronger than those of the single cis-eQTL per gene, even though individual cis-eQTLs can be quite strong.

5. How do the 210 gQTLs which do not colocalize with a trans-eQTL hotspot differ from the 398 gQTLs? Do they differ in effect size, direction, or the impacted conditions?

The table below shows a comparison between gQTLs that overlap a hotspot and those that do not:

	Value for gQTLs at hotspots	Value for gQTLs not at hotspots	Statistical test and significance

Absolute gQTL effect size	Median = 0.16	Median = 0.17	Wilcoxon p-value = 0.017
Direction of gQTL effects	Number of gQTLs in  • Plus = 201 • Minus = 197 	Number of gQTLs in  • Plus = 96 • Minus = 97 	Fisher's test OR = 0.97 P value = 0.93

The table shows that gQTLs that did not overlap a hotspot tended to have larger effects, but only by a small amount. There was no difference in the direction of these QTLs (in the table, "plus" indicates larger colonies with the RM allele). Comparing the gQTLs across conditions did not reveal a noteworthy pattern. Traits tend to have a similar fraction of gQTLs at hotspots:

Reviewer Figure 16: Number of gQTLs in each condition that do and do not overlap at least one trans eQTL hotspot.

In the interest of brevity, and given that no striking result emerged from these analyses, we have decided not to include these results in the revised manuscript.

6. The authors cluster the traits into two main groups (named green and blue) based on the genetic correlations (page 19). In the green group, they find GO term enrichments for ribosome biogenesis for genes with positive correlations and cell cycle control and various stress responses 30

for negative correlations. The blue group shows a reverse trend. The authors then make a comparison with genes associated with faster and slower chemostat-based growth in a previous study. This is an interesting observation but the analysis seems a bit shallow and perhaps oversimplified. Overall, do these green and blue growth conditions target distinct pathways such that this grouping makes sense? How do these conditions cluster based on the segregant growth data alone? It appears that most conditions involve some sort of stress in the form of metal ion or chemical treatment, yet stress conditions appear in both the green and blue groups. For example,

cisplatin is in the blue group while 4-NQO is in the green group. As both are DNA damaging agents, it is surprising that these are not clustered together.

We agree that these GO enrichment results are quite rich, but have deliberately kept our discussion of them at a fairly broad level, with the goal of focusing on the most salient patterns that arose from the data. We do think this is appropriate for the present paper, which already covers a fair bit of ground.

To address the reviewer's specific questions:

Supplementary Figure S1 showed correlations among growth traits based on individual segregant data. We have now added a version of this heatmap in which the traits are clustered by the correlations (Supplementary Figure S1B). The results show a range of clusters. For example, a prominent cluster is formed by various carbon sources (upper left corner of the heatmap, e.g., trehalose, sorbitol, lactate, lactose,...) with positively correlated growth. Mirroring these shared growth patterns, these conditions are all in the "green" group. The figure also shows that growth in YPD at 37°C (i.e., elevated temperature compared to the laboratory standard of 30°C) is anticorrelated with growth on these carbon sources as well as with growth at cooler temperatures (4°C and 15°C). These patterns are also reflected in Figure 6, where GO enrichments for growth at 37°C are often opposite to those in these carbon sources and at cooler temperatures. These results seem to "make sense" to us at least for these traits, and also show broad consistency between clustering based on phenotypes alone versus based on GO enrichments of the genetic correlations. Further dissection of how each trait is affected by genetic variation and how its causal variants intersect with gene expression effects will be conducted in future work.

Growth on the specific pair of conditions the reviewer mentioned (4-NQO and cisplatin) was not correlated ($r = 0.04$, $p = 0.24$), consistent with them being placed in different "groups" in this analysis. The reason for this may be different modes of action for these two compounds, where 4-NQO primarily causes bulky adducts repaired by nucleotide excision repair (Spasskaya et al., 2023), while cisplatin causes crosslinks repaired by both nucleotide excision repair and homologous recombination (Basu & Krishnamurthy, 2010; Silva et al., 2019). These differences in the mode of action of these two compounds may lead to differences in cellular responses, tolerance to stress and as a result, growth outcomes.

Reviewer Figure 17 (now included as Supplementary Figure 1B): Clustering of 46 growth traits based on the correlation of growth of 979 segregants in the respective conditions

Minor comments:

7. The manuscript could benefit from some streamlining. There are many redundant points made in the intro, results and discussion. For example, it is unnecessary for the authors to repeatedly mention that the eQTL and gQTL studies were done at different research institutions.

32

We agree and have removed the second enumeration of differences between studies from the first paragraph of the Results.

8. It is unclear what point the authors are trying to make in the results section titled "Most growth QTLs are colocalized with multiple local eQTLs". First they mention that "all but one of the 591 gQTLs in our dataset overlapped with at least one local eQTL, slightly but significantly more than expected by chance". To place this into context, it would help if the authors stated this number (i.e. the number of gQTLs that would be expected to overlap local eQTL by chance).

The expected number is 586, as reported in the paper. The complete sentence the reviewer

refers to was "All but one of the 591 gQTLs in our dataset overlapped with at least one local eQTL, slightly but significantly more than expected by chance (median of 1,000 random gQTL sets: 586, $p = 0.023$)."
(highlights added). As this information was already present, we have made no changes to the text.

9. It could be helpful to clear up the numbers for the eQTLs and the transcripts they impact. Fig 1 states that 5648/5720 genes have at least one eQTL. On line 108, the authors state "5,643 genes that have at least one eQTL in liquid YNB medium". Do the authors mean that 5,643 transcripts (or 5,648 - need to clear up what the exact number is), which is essentially all transcripts, have their levels impacted by an eQTL somewhere in the genome? Or do they mean that 5,643 genes have eQTLs mapped down at the gene-level, meaning that nearly every gene impacts the levels of a transcript somewhere in the genome?

The reviewer's first interpretation of line 108 is correct: there are 5,643 of genes whose expression is affected by at least one eQTL anywhere in the genome. We have clarified this in the text, which now reads "[...] we tested for correlations between the expression of each of 5,643 genes that are affected by at least one eQTL [...]"

Thanks also for spotting the discrepancy in the reported number of genes with an eQTL, which has been fixed in the Figure 1.

To avoid any potential confusion between the terms "gene" and "transcript" (which are equivalent in these data from *S. cerevisiae*, a species where alternative splicing is rare and was not considered here nor in Albert & Bloom et al, 2018), we have also changed the term "5720 transcripts" to "5720 expressed genes" in Figure 1A.

10. Fig 7B. It is difficult to distinguish between the red and gray circles due to the dense overlap. Perhaps these data can be shown in a different way to better separate Msn2 targets from non-targets, such as jittering the points or faceting the plot.

33

We agree with this concern, and have changed Figure 7B to separate the points indicated as Msn2 targets and non-targets by separately computing the cumulative fraction of mediating genes for the two categories.

11. The concept that "inherited alleles shape an individual's gene expression profile in a baseline condition in ways that can influence growth when cells encounter additional environmental factors" has been previously explored (e.g. see Gagneur et al. 2013, PMC3778020) This previous study showed that expression variation that is relevant to a phenotype tends to be persistent across different conditions. While the current study only looked at eQTLs in a single condition (YNB), this is worth mentioning in the discussion. This is also relevant to this sentence on line 127, page 5. "These effect sizes suggest that genetic effects on gene expression in a standard environment shape growth in different environments partially but not completely, resembling recent results in humans."

We thank the reviewer for this excellent suggestion and now cite this paper in the Discussion (page 26, line 419; page 28 line 478;) as well as the Introduction (page 4, line 95).

Referees' report, second round of review

Reviewer #1: The authors have thoroughly and satisfactorily addressed my comments.

Reviewer #2: The authors have addressed my concerns.

Reviewer #3: The authors have adequately addressed all concerns and made appropriate changes to the figures and main text. The authors have also performed considerable additional analyses to explore points raised by each reviewer. These have resulted in interesting new findings, for example identification of hotspots likely causal for specific traits (bottom page 13, main text), and resulted in numerous additional supplementary figures that strengthen the conclusions. Overall, the paper is substantially improved and should be of broad interest.

Authors' response to the second round of review

There were no reviewer comments to address.